# Non-Invasive Methodological Approach to Detect and Characterize High-Risk Sinkholes in Urban Cover Evaporite Karst: Integrated Reflection Seismics, PS-InSAR, Leveling, 3D-GPR and Ancillary Data. A NE Italian Case Study

**Alice Busetti [1], Chiara Calligaris [1],\*, Emanuele Forte [1], Giulia Areggi [1], Arianna Mocnik [2] and Luca Zini [1]**

[1] Mathematical and Geosciences Department, University of Trieste, Via Weiss 2, 34128 Trieste, Italy; abusetti@units.it (A.B.); eforte@units.it (E.F.); giulia.areggi@phd.units.it (G.A.); zini@units.it (L.Z.)

[2] Esplora Srl, Spin-Off University of Trieste, Via Weiss 1, 34128 Trieste, Italy; arianna@esplorasrl.it

\* Correspondence: calligar@units.it

**Abstract:** Sinkholes linked to cover evaporite karst in urban environments still represent a challenge in terms of their clear identification and mapping considering the rehash and man-made structures. In the present research, we have proposed and tested a methodology to identify the subsiding features through an integrated and non-invasive multi-scale approach combining seismic reflection, PS-InSAR (PSI), leveling and full 3D Ground Penetrating Radar (GPR), and thus overpassing the limits of each method. The analysis was conducted in a small village in the Alta Val Tagliamento Valley (Friuli Venezia Giulia region, NE Italy). Here, sinkholes have been reported for a long time as well as the hazards linked to their presence. Within past years, several houses have been demolished and at present many of them are damaged. The PSI investigation allowed the identification of an area with higher vertical velocities; seismic reflection imagined the covered karst bedrock, identifying three depocenters; leveling data presented a downward displacement comparable with PSI results; 3D GPR, applied here for the first time in the study and characterization of sinkholes, defined shallow sinking features. Combining all the obtained results with accurate field observations, we identified and mapped the highest vulnerable zone.

**Keywords:** sinkhole; PSI; 3D-GPR; reflection seismics; leveling; evaporites; geo-hazard

## 1. Introduction

Several European regions are affected by ground subsidence phenomena due to the presence of highly soluble evaporite rocks. The dissolution of soluble rocks and deposits at the surface, or in the subsurface, combined with internal erosion and deformational processes can produce depressions named sinkholes or dolines that represent a severe geo-hazard, as has occurred, among others, in France [1], Germany [2], Lithuania [3], Russia [4], Spain [5,6], United Kingdom [7,8], Albania [9], USA [10], South Africa [11] and Italy [12–15].

Sinkholes mainly occur when the bedrock is a soluble rock, mainly carbonates and/or evaporites and in terms of geohazard, there are noteworthy differences between lithologies [16,17]. In fact, gypsum and halite have much lower mechanical strength and more ductile rheology than most carbonate rocks, and their solubility is definitively higher, so in evaporitic terrains, the process is typically much faster and sinkhole frequency is commonly higher ([18] and the references within). These features

can reach depths of tens of meters and diameters of more than hundreds of meters. The largest known collapse sinkhole is Xiaozhai Tiankeng in China, 662 m deep and reaching 119.3 million m$^3$ in volume [19]. Crveno Jezero in Croatia has a vertical extension of 528 m, including the 280 m deep lake at its bottom [20]. The sinkhole discovered in Malta is a circular to elliptical collapse structure up to 600 m in diameter and with an average area of 56,000 m$^2$ [21]. Italy is also affected by this type of geohazard [22–27], and the results of recent investigations have shown that Friuli Venezia Giulia region (here after noted as FVG) is one of the most affected areas in northern Italy [18] with 1199 sinkhole phenomena inventoried just considering the evaporate bedrocks [15]. Here, sinkholes are well-known natural phenomena and have been recognized since the end of the 1800s [18,28,29], as they represent a serious threat to man-made structures, such as buildings and roads [14].

　　Gypsum crops out in almost all Italian regions, over about 1% of the total national territory [13]. Since most of these evaporite outcrops are very small, more or less detailed studies on gypsum karst have been carried out in only a few regions as Piedmont, Friuli Venezia Giulia, Emilia-Romagna, Calabria, and Sicily. In detail, karst areas in FVG are very common [14,30–35], but only 1% of the karstifiable lithologies are represented by evaporites. These crop out along the Alta Val Tagliamento valley and along the northern alignment, encompassing Pesarina, Pontaiba and Lumiei valleys. The Alta Val Tagliamento valley (Figure 1) is the most affected with hundreds of inventoried sinkholes [14]. Here, the combination of an intensively karstified evaporite bedrock, the presence of regional thrust, the high amount of annual rainfall (1600–2000 mm/y) and the large fluctuations of the water table (often greater than 10 m) seem to be responsible for the above-mentioned phenomena. One of the most notable sinkhole areas in the FVG is located in Quinis, a hamlet of the Enemonzo municipality (Figure 1).

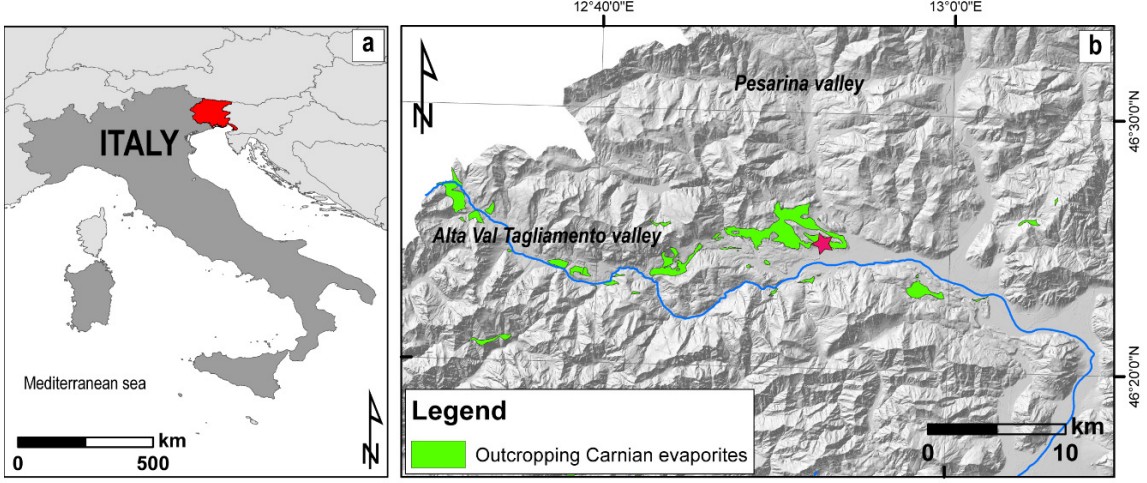

**Figure 1.** (**a**) Study area location. In red, the Friuli Venezia Giulia region (FVG); (**b**) focus on the Carnian evaporite outcrops (in green) at the NW side of the FVG, mainly within the Alta Val Tagliamento valley. In blue, the Tagliamento River. The red star is placed on Quinis hamlet.

　　Even in a small place such as Quinis, the identification and precise mapping of sinkholes are difficult tasks. The detection and adequate characterization of sinkholes and potentially unstable ground commonly require the application of multiple surface and subsurface investigation methods. Gutiérrez et al. [36] have described some of them: aerial and satellite images, topographic maps, field surveys, paleokarst analysis, subsidence damage maps, light detecting and ranging (LIDAR), synthetic aperture radar interferometry (InSAR), microseismicity, ground-based monitoring, hydrochemical modeling, speleological explorations, geophysical surveys, probing and drilling and trenching.

　　The most innovative methodological advances developed in recent years are related to the acquisition of interferometric SAR (InSAR) data that measure the displacement of the ground surface

using the phase difference between different radar acquisitions. In particular, the use of time series InSAR methods such as Persistent Scatterer Interferometry (PSI) provides deformation time series maps including retrospective analyses.

Remote sensing techniques contribute to the field of sinkhole hazard assessment by providing tools to implement sinkhole inventories and lending themselves to the monitoring of precursory deformations prior to sinkhole development [37].

These investigation techniques have been applied in various regions across the world (Figure 2). In general, they are particularly efficient in urban areas where Persistent Scatterers (PSs) can be easily recognized. However, PSs by themselves cannot be considered sufficient to outline subsidence areas, especially at large scales. Moreover, the effects of human activities and the low density of man-made buildings and infrastructures (i.e., vegetated areas) can significantly affect the final results. In these cases, the integration with different techniques such as field surveys, geophysical investigations and topographic monitoring is required to properly identify and monitor the phenomena. Topographic analyses are the most suitable techniques to investigate small building and road movements with time, while Ground Penetrating Radar (GPR) is one of the most efficient and high resolution geophysical techniques to image the subsurface at different depths and with various detail levels [38–41]. In fact, GPR frequencies can be tailored as a function of the expected targets, with high spectral components being able to reach centimetric resolution with a limited depth range while, on the contrary, lower ones can penetrate deeper but the output has a lower overall resolution [38]. In the framework of the research presented in this paper, we collected a full 3D GPR survey (see Section 3.4 within the Materials and Methods section) which, to the best of our knowledge, represents the first example of an application reported in literature for sinkhole characterization, while 2D GPR profiles are quite common e.g., [6,18,42,43].

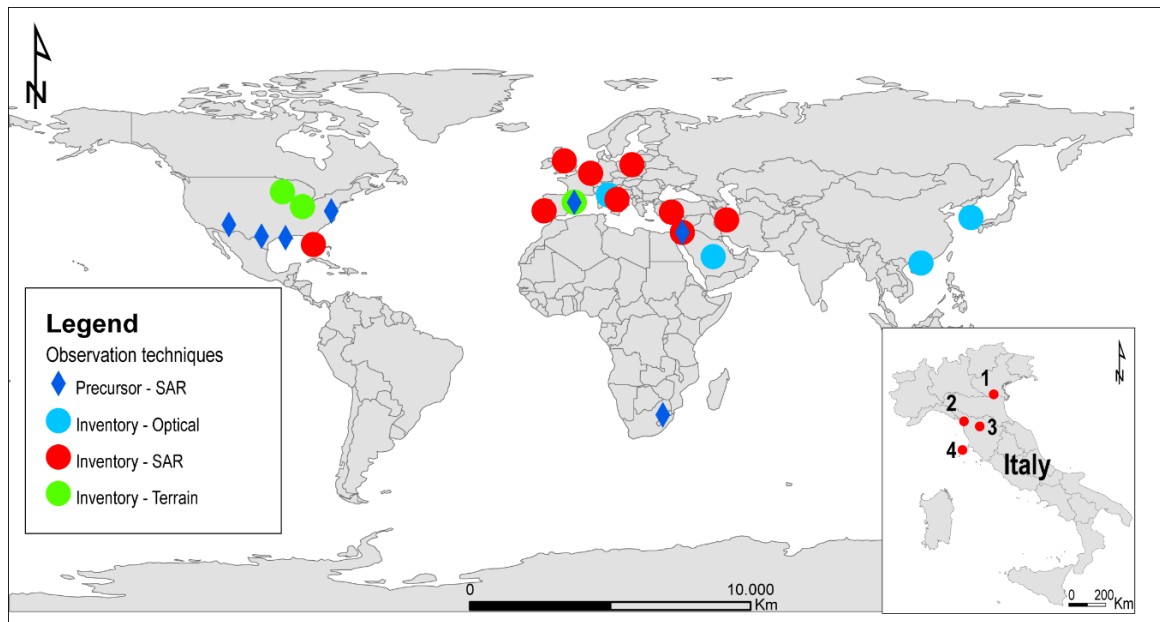

**Figure 2.** Observation techniques have been used worldwide for sinkhole precursor detection (blue rhombus) and sinkhole inventory using optical remote sensing (blue circles), remotely generated terrain models (green circles) and SAR techniques (red circles) [44–59]. The figure has been updated and modified only concerning the SAR part after Theron et al. 2018 [37]. In the box, SAR technique applied in the Italian territory to detect subsidence phenomena is shown: (1) Rovigo (Veneto region); (2) Camaiore (Tuscany region); (3) Prato (Tuscany region); (4) Elba island (Tuscany region).

In the present paper, the authors analyze the Alta Val Tagliamento Valley, focusing on the Quinis area where sinkholes have been reported since the beginning of the last century [60].

The valley is a good example of the geological context typical of mountain areas where an evaporitic bedrock is mantled by Quaternary deposits with variable thicknesses. This location was chosen as a test site to apply different techniques in order to better outline the most hazardous areas and also integrate PSI, which is applied for the first time to this topic in Italy in the evaporitic context, integrated with topographic leveling and integrated reflection seismic and 3D GPR, which also represents a novelty in such kind of investigations.

## 2. Study Area

The study area is located in the mountain sector of FVG, NE Italy, in the Alta Val Tagliamento Valley within the Enemonzo municipality, in the hamlet of Quinis. Some findings testify that Enemonzo is certainly one of the most ancient settlements and it has suffered from the presence of sinkholes for a long time. The occurrence of these phenomena is favored by the presence of Triassic evaporites, Carnian in age, in the entire valley floor (Figure 1). The evaporitic bedrock (Figure 3) does not outcrop extensively, it is mainly mantled by Quaternary deposits due to alluvial fan deposits that prograde over the fluvial terraces of the Tagliamento River (Figure 4). The evaporites go under the Raibl Formation and, according to the available literature [61] (Figure 4), are subdivided into three different members, which can be described as follows from the bottom to the top:

- *Red shales member* (RBA1) with a thickness between 80 and 100 m, characterized by red shales and siltstones typical of a fluvial environment close to the coast;
- *Gypsum and grey dolostones member* (RBA2) characterized by a thickness of 350 m; it is primarily composed by grey and white saccharoid gypsum with marl inclusions at the top (Figure 3), yellowish dolomitic marls, and to a lesser extent, blackish or greenish clays and dark limestone in thin layers;
- *Marls and dolostones member* (RBA3) with a thickness of 180 m; it is characterized by grey dolostones often vacuolar and cataclastic, marls and multicolor clays close the depositional sequence.

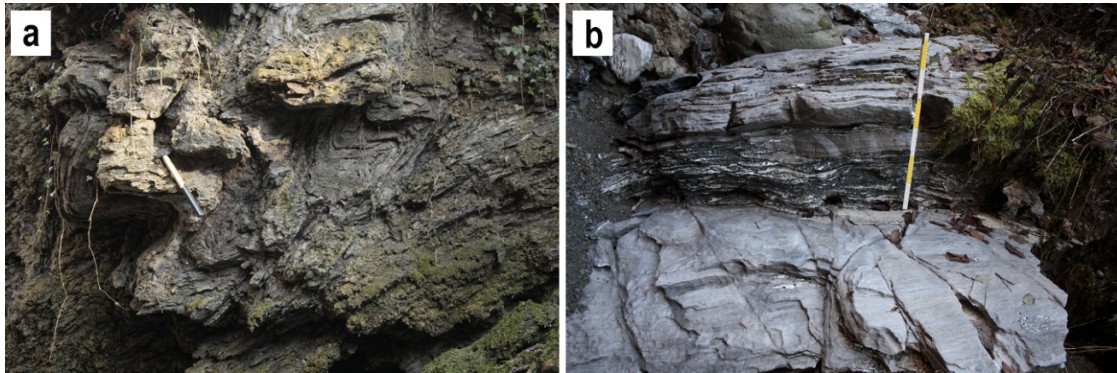

**Figure 3.** Evaporites outcrop in the hilly areas north of Quinis village. (**a**) Small folds highlighting the plasticity of the lithology; (**b**) intercalations of thin layered gypsum within carbonate clayey silt.

Thanks to the 25 boreholes that were drilled between 2005 and 2013, and to the previously acquired geophysical data, a conceptual model of the area has been already hypothesized [18]. It highlights the presence of Quaternary deposits as glacial till, alluvial and colluvial levels over an evaporitic bedrock. The complexity of the depositional pattern reflects on the heterogeneity. The thickness is variable from north to south of the study area (Figure 4). To the North, it is approximately a few meters and deepens moving towards the South up to more than 60 m in correspondence with the Tagliamento River terraces. The alluvial deposits consist of highly permeable polygenic gravels, which include lenses of less pervious clay and clayey silt. Although the gravels are locally cemented, the geotechnical characteristics of the whole deposit are quite poor, with a high vertical and horizontal variability of the bearing capacity. In the most part of the boreholes, SPT (Standard Penetration Test) investigations have

been realized in correspondence with clay and clayey silt lenses. The SPT results go from a minimum of n SPT equal to 0 up to 18 with some voids up to 2 m wide identified while realizing the tests.

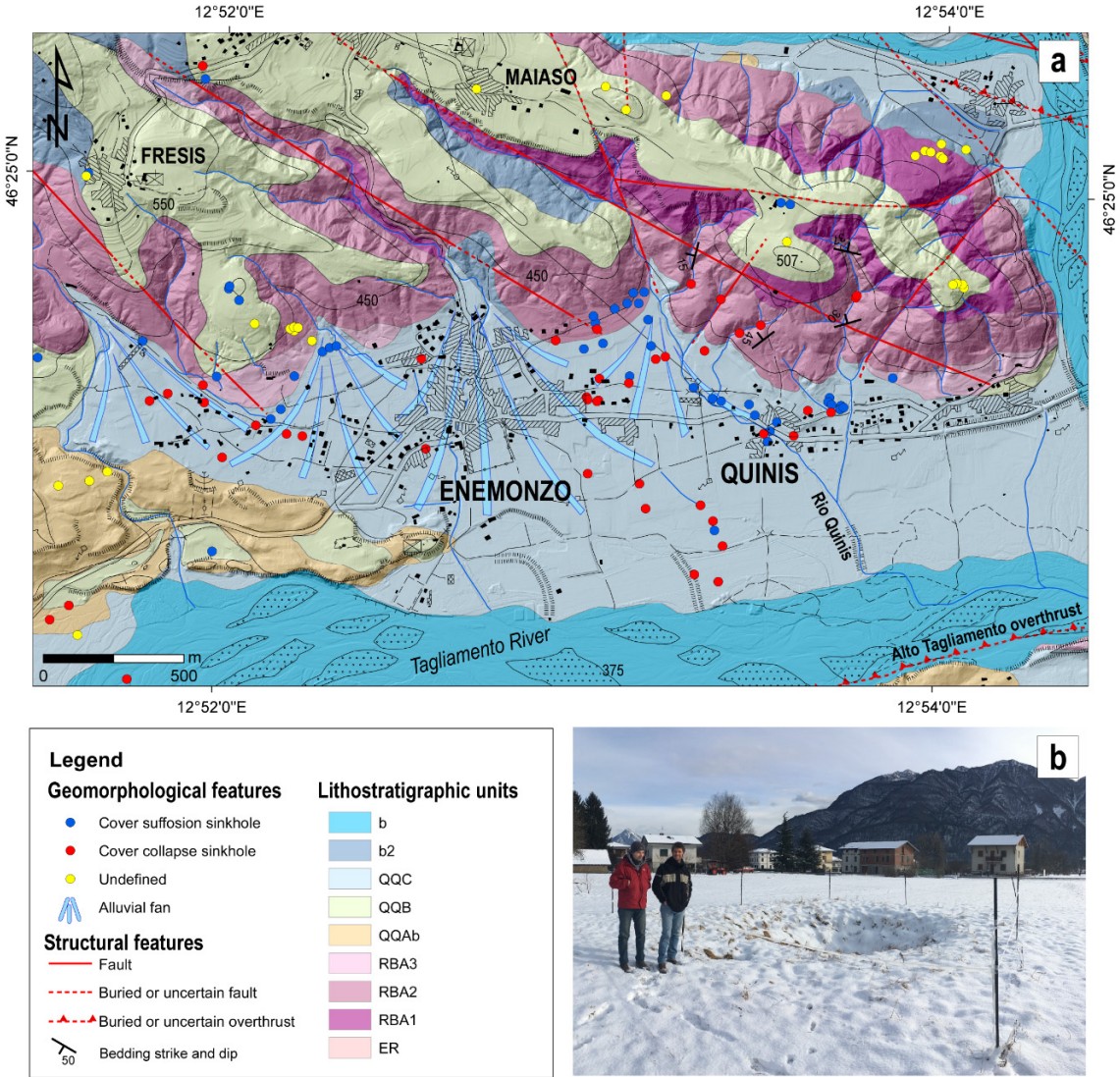

**Figure 4.** (**a**) Geological and geomorphological map of the study area (modified after [61]). Lithostratigraphic units: (**b**) actual alluvial deposits; (b2) eluvial-colluvial deposits; (QQC) fluvial gravel and sand; (QQB) glacial till; (QQAb) fluvial conglomerates. All these units are Pleistocene–Holocene in age. (RBA3) marls and dolostones member; (RBA2) gypsum and grey dolostones member; (RBA1) red shales member. RBA1, RBA2 and RBA3 belong to the Raibl Formation, Triassic in age. (ER) Siera group (Triassic age), mainly limestones and dolomitic limestones badly stratified, mainly massive. (**b**) One of the cover collapse sinkholes present in the area initially formed in 1977. It reactivated six months after the first collapse and in 2012. In 1985, few meters from where the phenomenon previously occurred, a new sinkhole appeared. Photo in (**b**) was taken in 2013.

From a structural point of view, the study area is characterized by the presence of several faults that are NW–SE oriented [62]. The EW Tagliamento Valley is controlled by the regional Alto Tagliamento line. This overthrust separates the "Alpi Tolmezzine" (northern sector) from the "Prealpi Carniche" (southern sector), resulting in the Raibl Formation in the hanging-wall overthrusting the Monticello Formation (in the S, out of the geological map presented in Figure 4). This structure, buried in the study area and recognizable only to the West, dips approximately 60° to NW [61].

The Quinis village area is characterized by the presence of an extensive phreatic aquifer with relevant water table fluctuations, fed by two contributions: the effective infiltrations from one side and the stream and river leakages from the other side. Stream leakages mainly influence the northern sector, while the Quinis torrent crosses the investigated area in the NE–SW direction. Tagliamento River leakages are instead more important in the southern portion of the area [15,18].

The groundwater flow is conditioned by the structural setting of the evaporitic bedrock and by the extreme heterogeneity of the Quaternary deposits. The aquifer system is very complex, and the water table has a rapid response to rainfall, in particular the water level fluctuations are large, with oscillations between 6 and 32 m. One of the main rainfall events recorded 248 mm of rain in 56 h (24–26 December 2013) with a consequent water table rise of 10 m and a maximum recorded velocity of 40 cm/h [18], witnessing a fast circulation of the groundwaters. This hydrogeological context and especially the wide and fast water table fluctuations have produced high karstification of the gypsum and grey dolostones member (RBA2).

Calligaris et al. [63] described the field experiment of placing rock evaporitic samples into piezometric tubes at different depths, demonstrating a dissolution rate of the evaporites up to 2.8 mm/y, almost three times greater than that expected if compared with the available literature data ([63] and in text references).

In between 1947–1948, Cosano [60], supervised by Professor. A. Desio in his thesis, described the area of Enemonzo-Quinis, observing its geostatical instability due to the presence of chalks in the bedrock.

Gortani, in 1965 [28], described an important cover collapse sinkhole that occurred in 1964 close to the Tagliamento River. After the catastrophic event in 1964, there has been an increment of the phenomena. Actually, six houses have been relocated due to their important damages. On the whole, in the municipality, there are 208 inventoried phenomena, of which 46 are cover suffosion sinkholes, 40 are cover collapse sinkholes and the remaining sinkholes have an undefined typology. They range in size from few tens of cm up to 75 m, reaching depths that are in the range from a few centimeters up to more than 15 m. Focusing just on the Quinis area, 32 phenomena were recognized (Figure 4), some of them are actually quiescent, but some others are active and continue to evolve, causing important damages to the infrastructure (Figure 5).

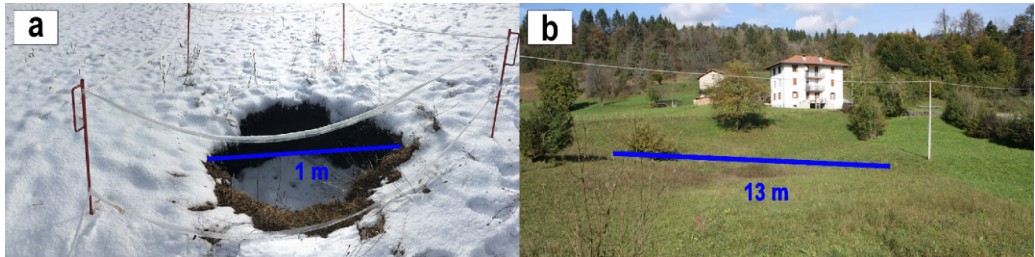

**Figure 5.** Two sinkholes recognized in the area. (**a**) Cover collapse sinkhole occurred close to the Rio Quinis riverbed. The hole, after the occurrence, has been promptly replenished from the land owner with loose material (about 2 m$^3$) (photo acquired in 2017); (**b**) historical cover suffusion sinkhole, which transforms to an ephemeral lake when it rains (photo taken in 2015).

## 3. Materials and Methods

Here, we individually introduce the methods adopted in this study, ideally with increasing resolution level, but we remark that they have to be compared, combined and integrated in order to obtain more constrained and cross-validated results.

### 3.1. Reflection Seismics

Reflection seismics is a mature geophysical technique applied worldwide both onshore and offshore for many different applications and at various scales. One of the most important objectives is

finding and characterizing hydrocarbon reservoirs [64]. The general principle is based on sending artificially generated seismic waves into the subsurface where the different structures and lithologies reflect back a portion of the transmitted energy as a function of the acoustic impedance contrasts. The reflected energy is then recorded at the surface by dedicated sensors having 2D or 3D geometries. Recorded data have to be properly processed and analyzed, often with sophisticated algorithms [65], obtaining a visual representation of the geological structures down to the depth of kilometers and reaching a resolution of tens of meters. Ultra-high-resolution surveys are also possible for shallower depths and for engineering or shallow geology studies. Here, we remark that reflection seismics is totally different from refraction seismics, as far as their physical bases, the data processing, and the expected results. Reflection seismics have been used for a long time for sinkhole-related issues, usually to give a general overview of the geology of the area prone to subsidence. In particular, there are examples focusing on the failure of anthropic infrastructures [66], on ground deformations induced by mining activities [67] and on sinking in urban areas [68], as well as exploiting shear waves [69].

In this paper, we collected some 2D reflection seismic profiles by using a Geode (Geometrics) seismograph connected to up to 48 channels in order to get information in the first hundred meters below the topographic surface. As a seismic source, we used a 5 kg sledgehammer, which provided enough energy for the objective of the work. After some preliminary tests (walkaway test) to evaluate the best acquisition geometry and parameters, we set the survey parameters as reported in Table 1.

**Table 1.** Acquisition parameters of the reflection seismics' profiles.

| Channel Number | *up to 48* |
| --- | --- |
| Channel distance [m] | *2* |
| Shot interval [m] | *2* |
| Minimum Offset [m] | *1* |
| Maximum Offset [m] | *94* |
| Vertical stacking | *4* |
| Trace length [s] | *1* |
| Sampling interval [ms] | *0.25* |

Both geophones and shot point positions have been recorded by using an RTK GPS, obtaining centimetric and decimetric accuracy in latitude/longitude, and elevation, respectively. Processing flow is summarized in Table 2. Additional algorithms (coherency filters and deconvolution) have been applied on specific subsets of the data characterized by poor overall continuity and low signal-to-noise ratio. Migration was not applied due to the limited dip of seismic reflectors and because diffractions were used as an additional aid to highlight lateral variations. In order to improve the seismic velocity field reliability, we applied, in addition to the standard velocity analysis, the common reflection surface (CRS) approach. The whole data processing and analysis were performed by using Seispace ProMAX suite (Landmark, Halliburton).

**Table 2.** Processing flow applied to reflection seismics' profiles.

| Data Editing |
| --- |
| Geometry assignment and sorting |
| Static corrections |
| Coherent noise (ground roll) attenuation |
| Amplitude analysis and recovery |
| CMP velocity analysis and NMO correction |
| Stacking/weighted stack |
| Depth conversion |

### 3.2. PS-InSAR

Interferometric data (PSI) provide useful information about the surface deformation due to natural and/or anthropogenic phenomena. By exploiting the measure of the signal phase change between two radar images acquired with the same geometry in different time periods, it is possible to obtain maps of the ground displacement over time with respect to the satellite itself in the line-of-sight (LOS) [70–72].

In particular, time series InSAR methods are able to measure and monitor displacement over a given period of time with high accuracy by processing multiple interferograms derived from a stack of radar images. Among the time series InSAR techniques, Persistent Scatterer Interferometry (PSI) focuses on point-like coherent targets dominated by a single scatterer [73]. The scatterers, characterized by high phase stability, can usually be detected in urban areas due to the presence of many stable points such as man-made structures. The PSI approach is based on the processing of many interferometric data pairs derived by a stack of single-complex look (SLC) radar images and a common master image.

In this paper, we used SAR data acquired by the Italian COSMO-SkyMed satellite, characterized by high resolution microwaves in X band (wavelength 3.1 cm) and provided by the geological survey of the FVG in the framework of different research agreements between Trieste University and the survey itself. The used datasets, 3a_A in ascending orbit and 03a_D in descending orbit, span the period 18 February 2012–8 September 2016 and 1 January 2012–8 September 2016, respectively.

Data were initially processed by Planetek Italia S.r.l. for the area of interest of selecting the dataset and the master image (28 April 2014 master chosen for the ascending geometry; 30 July 2013 master chosen for the descending geometry). Then, the Stable Point Interferometry over Un-urbanized Areas (SPINUA algorithm implemented in Rheticus®) algorithm was applied [74]. On the obtained elaborated datasets (co-registered amplitude stack and differential interferograms stack), the algorithm selected the position of pixels with high temporal consistency, known as persistent scatterers (PSs). After the removal of the atmospheric artifacts and the correction of Digital Elevation Model (DEM) errors and terrain motion, the outcomes consisted of PS maps (in shape file format) with their relative information about position and displacement along the line-of-sight (LOS) [74]. In the end, the elaborated data, for every PS, contain the following information:

- An average velocity value (mm/y) referring to the displacement in LOS over the entire period of observation;
- An average velocity value (mm/y) referring to the displacement in LOS over each year;
- An estimated value of the cumulative displacement (mm);
- Standard deviation of the velocity with a mean value of 0.27 mm/y for the ascending dataset and 0.19 mm/y for the descending one.

Since the SAR data provided information about terrain motion with respect to the satellite along its LOS, we computed the vertical and the E-W velocity components by using data acquired in both ascending and descending satellite orbits.

We assumed a zero north contribution because of the lower sensitivity of the satellite to detect motion in the north component due to the near-polar satellite orbit [75].

A grid of $15 \times 15$ m was initially defined and, for each cell, the average velocity of the PSs was computed in the ascending and descending datasets ($\Delta d_{asc}$ and $\Delta d_{desc}$). Then, considering the local incidence angle counted positive from the vertical ($\theta$), and the azimuth of the satellite heading (positive clockwise from north) ($\varphi$), we calculated the deformation in E-W and vertical components by using the following formula [75–77]:

$$
\begin{pmatrix} \Delta d_{asc} \\ \Delta d_{desc} \end{pmatrix} = \begin{pmatrix} -\cos\varphi_{asc}\sin\theta_{asc} & \sin\varphi_{asc}\sin\theta_{asc} & \cos\theta_{asc} \\ -\cos\varphi_{desc}\sin\theta_{desc} & \sin\varphi_{desc}\sin\theta_{desc} & \cos\theta_{desc} \end{pmatrix} \begin{pmatrix} \Delta_E \\ \Delta_U \end{pmatrix} \tag{1}
$$

The values of the $\varphi$ and $\theta$ angles are $-12°$ and $27°$ for ascending track and $-169°$ and $33°$ for descending track (as provided by Planetek).

### 3.3. Precise Geometric Leveling

In between October 2012 and August 2015, 9 leveling monitoring surveys (October 2012, January 2013, May 2013, August 2013, December 2013, April 2014, August 2014, December 2014, August 2015) on selected buildings were performed by a topographical technical study [78] using the DiNi precision level and the total station. Furthermore, 49 points were analyzed by placing 21 optical prisms (TA code) on the upper part of the buildings and 21 vertical landmarks (VB) code) at the base of the same buildings. As reference, 7 benchmarks were placed (initials v and cs in positions considered stable within the village (Figure 6).

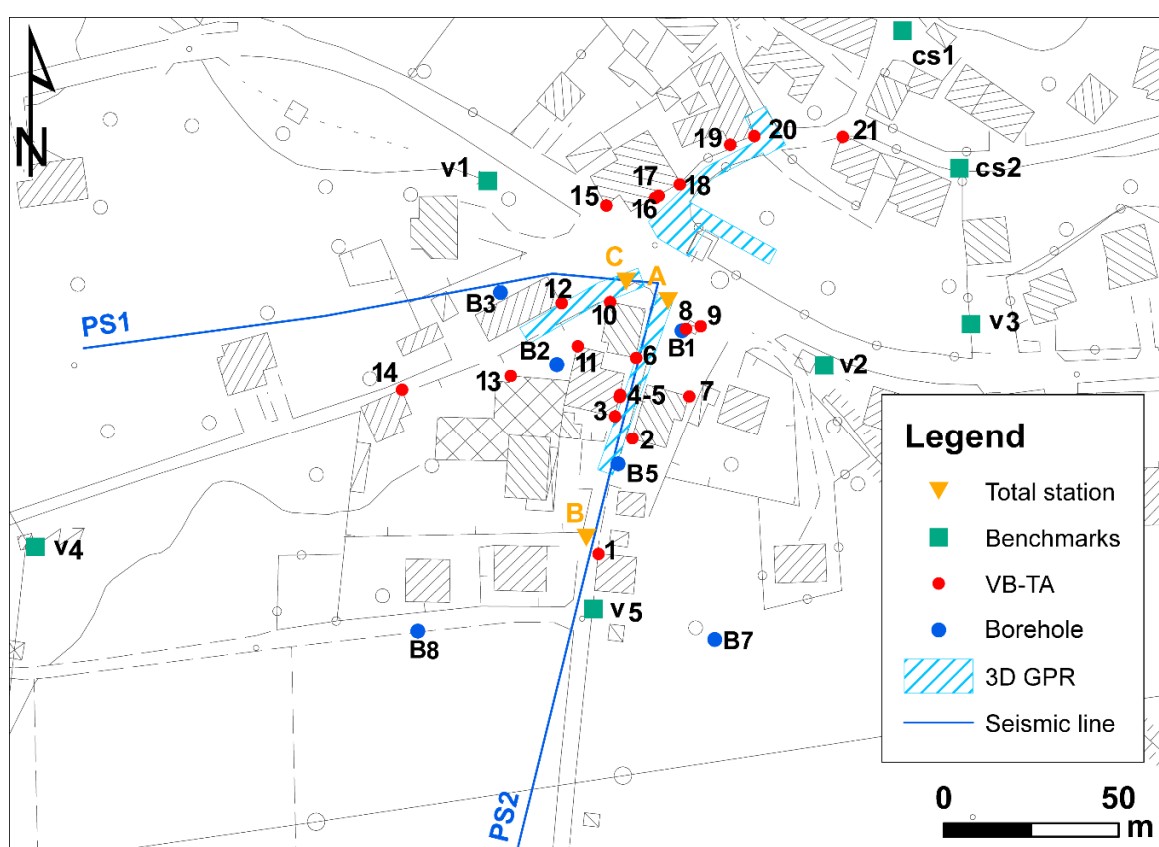

**Figure 6.** Location of direct and indirect measurements map: seismic lines (PS1 and PS2); benchmarks v1–v5 and cs1–cs2; B1–B8 boreholes; 1–21 leveling monitoring points; A, B, C total station monitored points; in pale blue—the area covered by the full 3D GPR survey.

In detail, precise geometric leveling measurements were carried out using the DiNi precision level with invar barcode stages, reaching an accuracy of 0.3 mm on all the vertical benchmarks (VB). The optical prisms (TA) are detected through total station measurements with TRIMBLE 5601and TRIMBLE S6 robotic total stations, having a precision equal to 2 and 3 mm, respectively.

Leveling data are used to define the vertical movements and are considered reliable to the tenth of a millimeter. The values obtained with the total station are not used for the computation of the quote. With the latter instrument, horizontal movements, along the coordinates E and N, are considered valid if greater than 2 mm.

### 3.4. 3D GPR

One of the main issues of GPR applications for sinkhole-related studies is the extreme spatial variability, which requires dense surveys. In the last few years, there is an array of GPR equipment on the market, which are able to collect actual 3D GPR data volumes faster e.g., [79,80]. In fact, with single channel (i.e., classical bistatic) GPR equipment, only 2.5D surveys can be collected [81]. Moreover, while until now the new 3D systems were applied mainly for archaeological applications [82,83], for pipe and tree root detections [84] and for pavement assessments [85], geological applications are less common and limited to the detection and imaging of faults and fractures [86,87]. Moreover, the last type of applications are actually very dense 2.5D surveys collected with traditional GPR equipment rather than with antenna arrays due to logistical constraints related to the large dimensions and heaviness of multichannel systems, which in turn make their application on rough surfaces difficult.

In this study, we used the 3D MiniMIRA array GPR (Malå Geoscience) equipped with 5 transmitting and 4 receiving 400 MHz shielded antennas, allowing the collection of 8 parallel profiles with a constant distance equal to 8 cm. In order to optimize the spatial resolution, we set a trace spacing also equal to 8 cm to obtain a constant in-line and crossline coverage. The system is connected with an electromechanical odometer for triggering and with RTK GPS for accurate absolute positioning. All the registered data are then combined in a single project and processed with the dedicated rSlicer software, as well as with an interpretation suite originally developed for reflection seismic data (Petrel, Schlumberger). By using these tools, it was also possible to calculate GPR attributes that further benefit from the full 3D geometry [88]. Details on the application of attributes to GPR data are beyond the purposes of this paper and can be found in [81,89].

## 4. Results

### 4.1. Reflection Seismics

The good overall quality of seismic data allowed the investigation down to a depth of at least 60 m, reaching a resolution of a few meters. Figures 7 and 8 report two almost perpendicular profiles, representative of the entire study area (Figure 6). The seismic interpretation was helped and validated by the stratigraphy of some boreholes drilled in the past in the area. In detail, the top of the evaporitic bedrock (TB) is clearly imaged along all profiles. It dips towards S with an irregular trend, locally showing abrupt changes in depth, such as, for instance, in correspondence with the white arrow in Figure 8. In the E-W profile (Figure 7), there are no clear trends but analyzing both the TB horizon and some other reflectors within the evaporites, we can highlight some depocenters characterized by local (about 20 m wide) deepening (white arrows). It is interesting to notice that, despite the intrinsic resolution limits of the method, both profiles show that the sinking is accommodated by the surface sedimentary layers, which are dipping less as they are closer to the topographic surface. By analyzing the P-wave interval velocities inferred from the integrated common midpoint (CMP) and common reflection surface (CRS) analyses (Figures 7b and 8b), the presence of the evaporites is clear due to their high velocity in comparison with the overlying sediments. Even more interesting is a local velocity inversion along the N-S profile (Figure 8), located where the sinking is more evident. This testifies that there are lenses and intercalations of materials with poor geotechnical properties, and thus further increasing the instability of the buildings and the overall risk of the area.

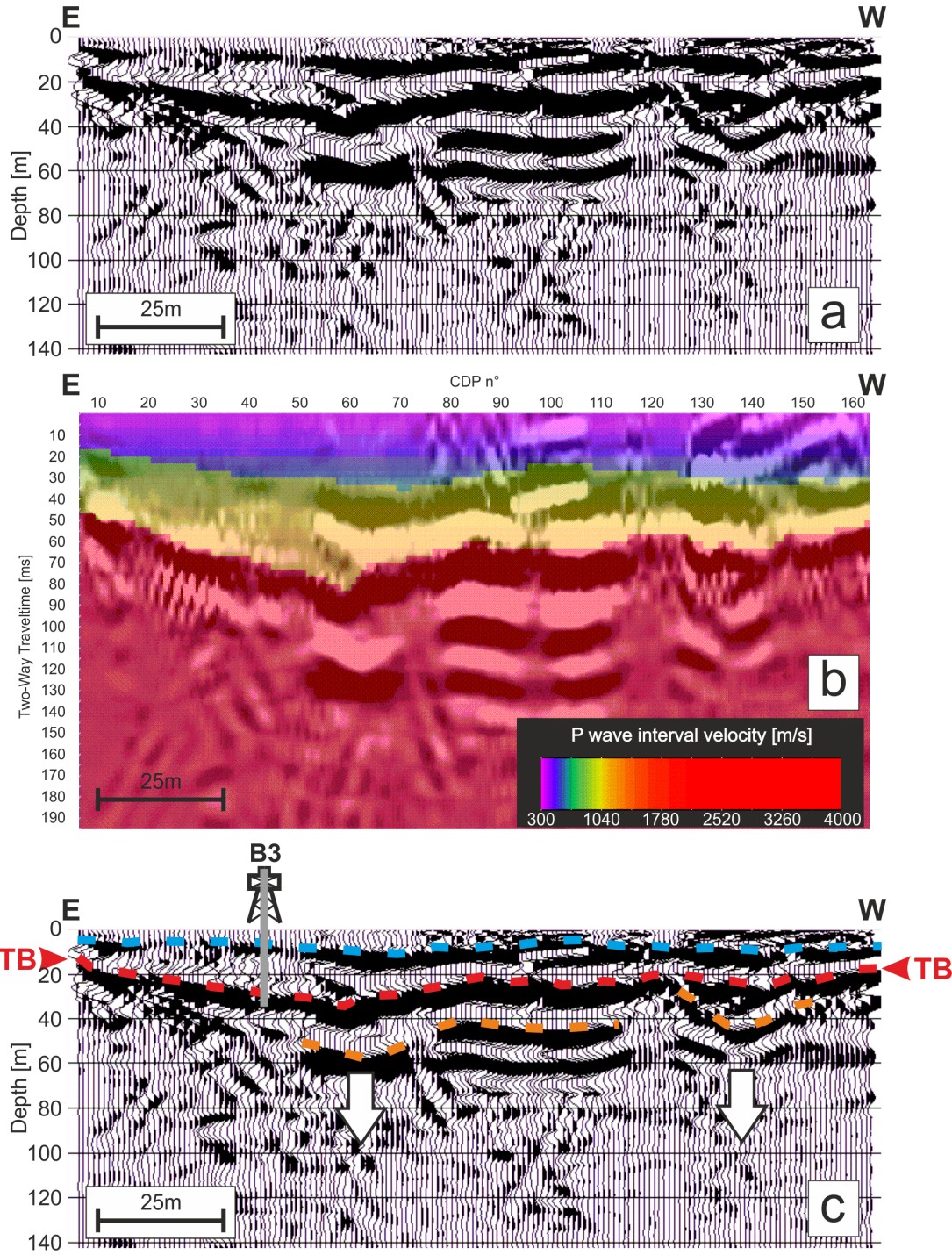

**Figure 7.** Reflection seismic profile PS1. (**a**) Processed depth converted profile; (**b**) Two-way traveltime (TWT) profile with the interval velocity superimposed; (**c**) interpreted depth converted profile. The red dashed line marks the top of the bedrock (TB), while light blue and orange lines highlight seismic horizons within the sediments and within the evaporites, respectively. The location and depth of borehole B3 used for validation are also provided.

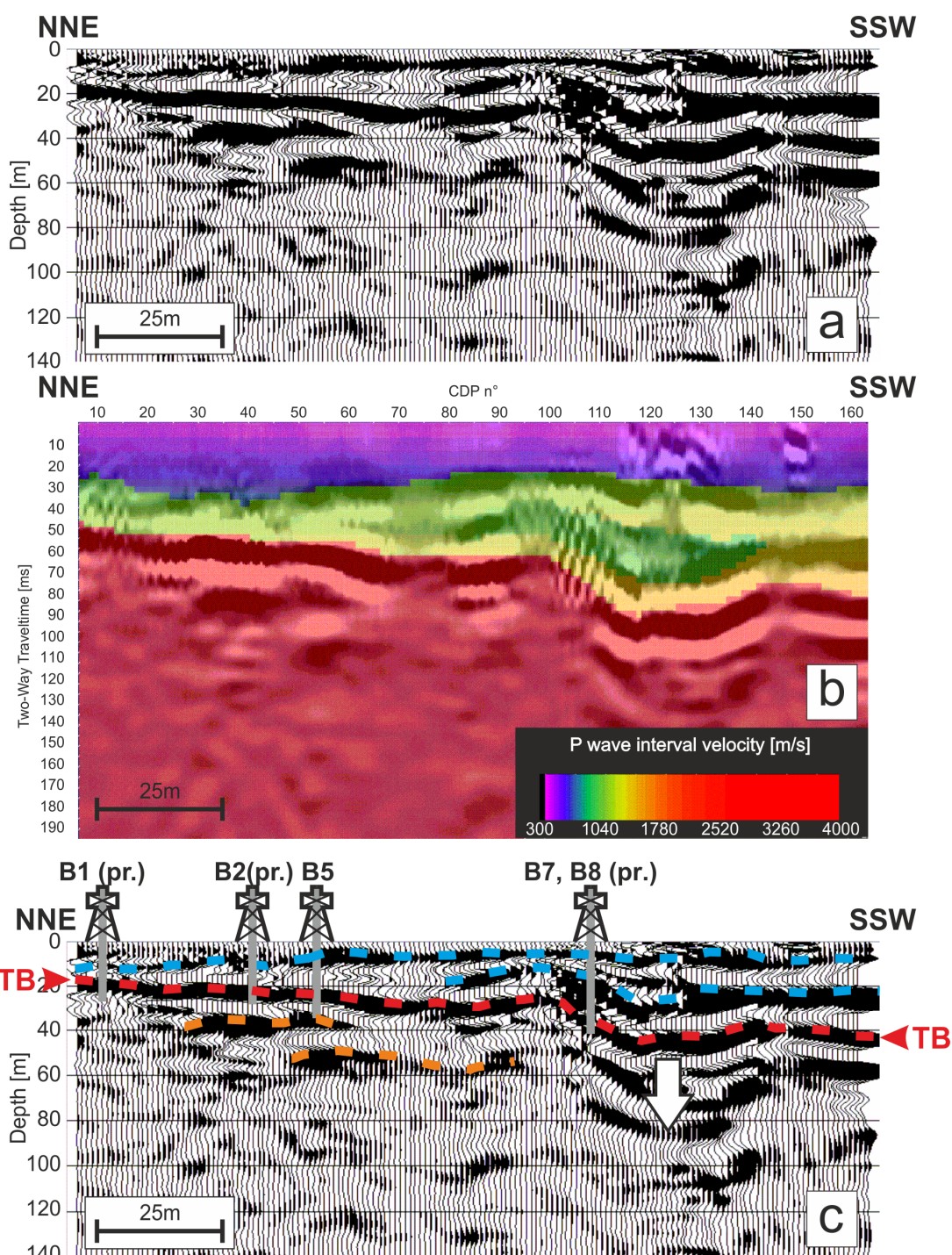

**Figure 8.** Reflection seismic profile PS1. (**a**) Processed depth converted profile; (**b**) TWT profile with the interval velocity superimposed; (**c**) interpreted depth converted profile. The red dashed line marks the top of the bedrock (TB), while light blue and orange lines highlight seismic horizons within the sediments and within the evaporites, respectively. While the arrow is located where the deformation reaches the maximum. The location and depth of boreholes B1, B2, B5, B7 and B8 used for validation are also provided (Figure 6).

## 4.2. PS-InSAR and Leveling

The analysis of a territory through a remote sensing approach allows the investigation of an area with a different perspective in addition to just the aerial view. In particular, PSI is considered to be one

of the most effective techniques to investigate subsiding areas based on the characteristics of the SAR instruments on the satellite. The satellite can acquire images during days and nights (active system) and in every meteorological condition thanks to the cloud-penetrating capability of the microwave band. The measure of changes in range distance of the target on the ground with respect to the satellite provides the measurement of centimetric/millimetric surface deformations of the terrain [72].

With this in mind, the interferometric data provided by the geological survey of the FVG region for the area of Enemonzo and Quinis have been analyzed. From both the ascending and descending datasets (Figure 9a,b), emerges an important mean displacement velocity especially in the Quinis hamlet, with values up to −6 mm/y. With the velocity values being negative in both geometries, we notice the presence of a vertical downward displacement, considering that a negative value represents a movement of the surface away from the satellite. This is not a surprise seeing that the area has been considered unstable since the early nineties when Cosano [60], in his master's degree thesis entitled "*Geological Analyses and Investigation of the Enemonzo Area (Carnia)*", highlighted the nature of the territory. Thanks to all the previous studies, it is understandable how the problem of subsidence in Quinis is historical, which goes beyond the construction methods of the buildings, the presence of heavy traffic on the state road and the activation of hydroelectric power plants along the Tagliamento River and its tributaries during the sixties, even if all these phenomena may have exacerbated the instability problems.

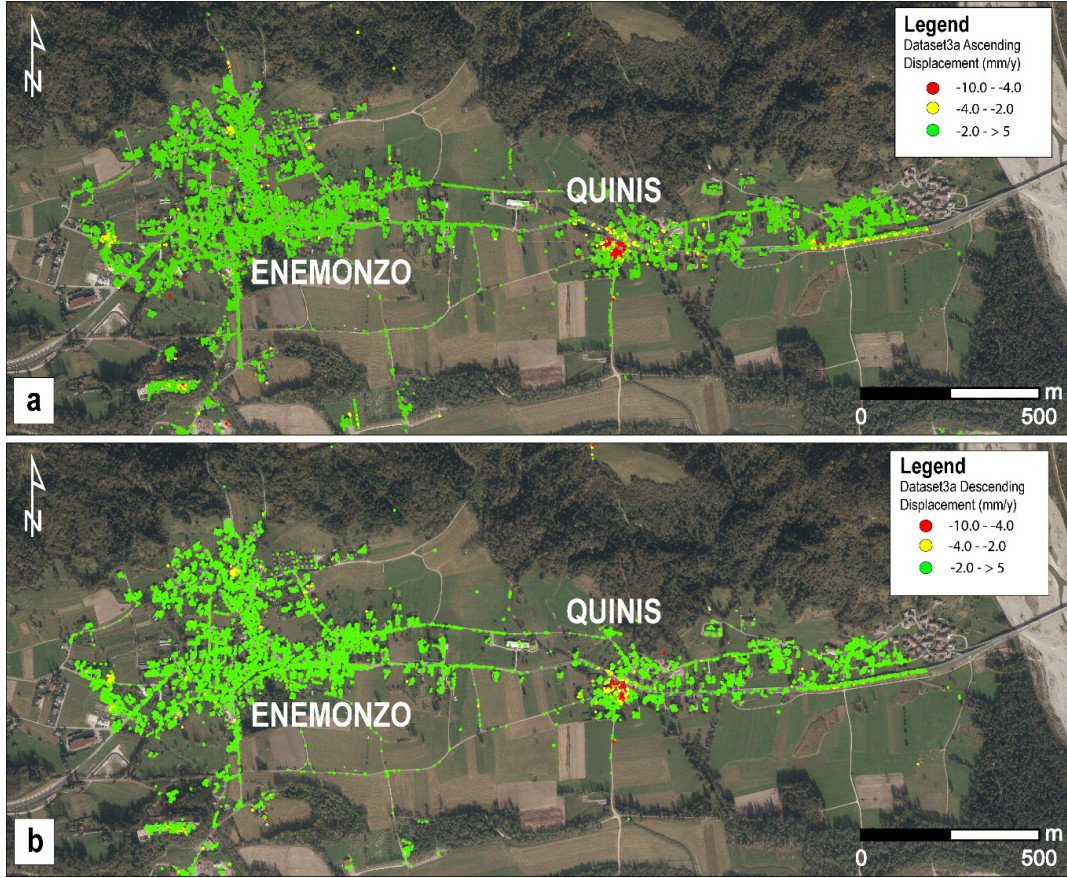

**Figure 9.** Average velocity of movement of the PS (mm/y) with reference to the entire observation period. The estimated velocity is projected along the line of sight of the satellite. (**a**) Ascending geometry (18 February 2012–08 September 2016); (**b**) descending geometry (01 January 2012–08 September 2016). In red, negative values highlighting high downward deformations; in green, positive values highlighting upward deformations and stable zones.

Despite the important studies funded initially by the civil defense and later by the geological survey of the FVG region, which has been realized in the area since 2009 by the researchers of the mathematics and geosciences department of Trieste University, it is actually difficult to outline the unstable areas. This is mainly due to the intrinsic lithological characteristics of the bedrock, which is heavily karstified and heterogeneous even in very close areas. In addition, even if we are in a mountainous area or in a small hamlet, the anthropic factor is important. Human actions tend to completely mask the geomorphological evidence of sinkhole activity. In sight of this, the interferometric approach, thanks to its coverage and high spatial resolution, represents a useful tool to investigate the study area, not only focusing in the center of Quinis where previous studies have highlighted the main criticisms, but enlarging the analysis to a wide portion of the Tagliamento Valley. To improve our investigation, we subdivided the area with grid size cells of 15 × 15 m. This value is surely larger than the achievable resolution, but it allows us to highlight the most significant and constrained subsidence zones. For each of them, we computed the average velocity in both orbit tracks (ascending and descending) and later, applying Equation (1), we calculated the average vertical and horizontal velocity components. Figure 10a summarizes the areas impacted by the vertical velocities, which are within the range from −6 up to +2 (mm/y).

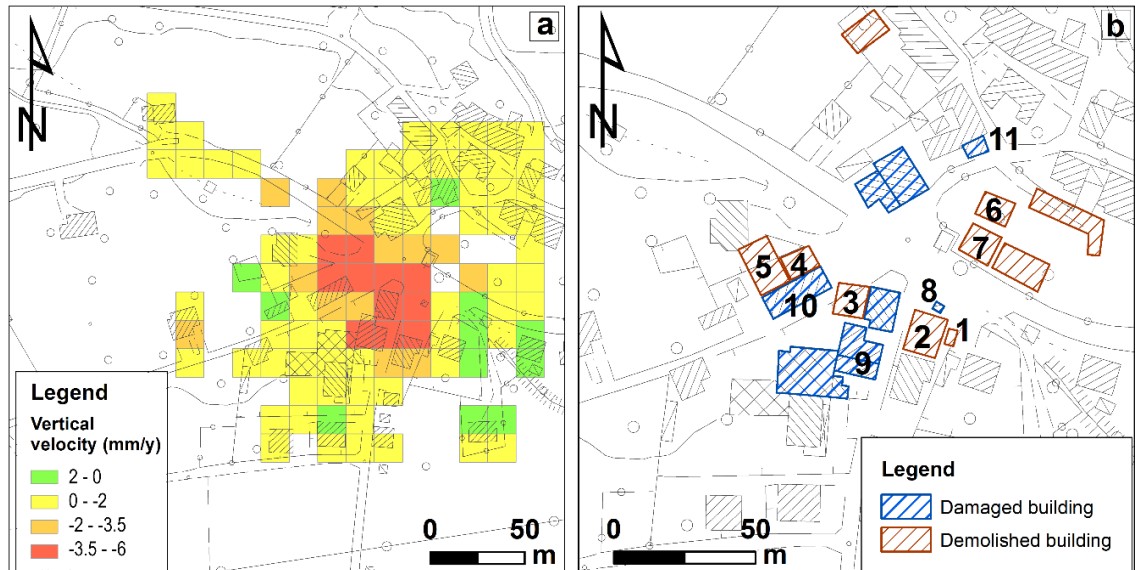

**Figure 10.** (**a**) Vertical velocities in mm/y computed using Equation (1) on the interferometric available data for the period 2012–2016; (**b**) demolished buildings (brown) and actually damaged houses (blue) in the central part of the study area. Numbers were assigned to the houses described further on in the text and in Figure 11.

Comparing the map with the demolished and damaged buildings highlighted in Figure 10b with the area where the downward vertical velocities are higher in Figure 10a, we can notice a very high correspondence. In fact, it is apparent that the buildings and infrastructures that, within recent years, had instability problems are concentrated in red/orange cells, and thus demonstrating that the present day deformations are mainly concentrated where they occurred in the recent past. Furthermore, we must consider that Quinis now appears different from the past in terms of the number of buildings, mostly as a consequence of the long term effects of subsidence and related damages.

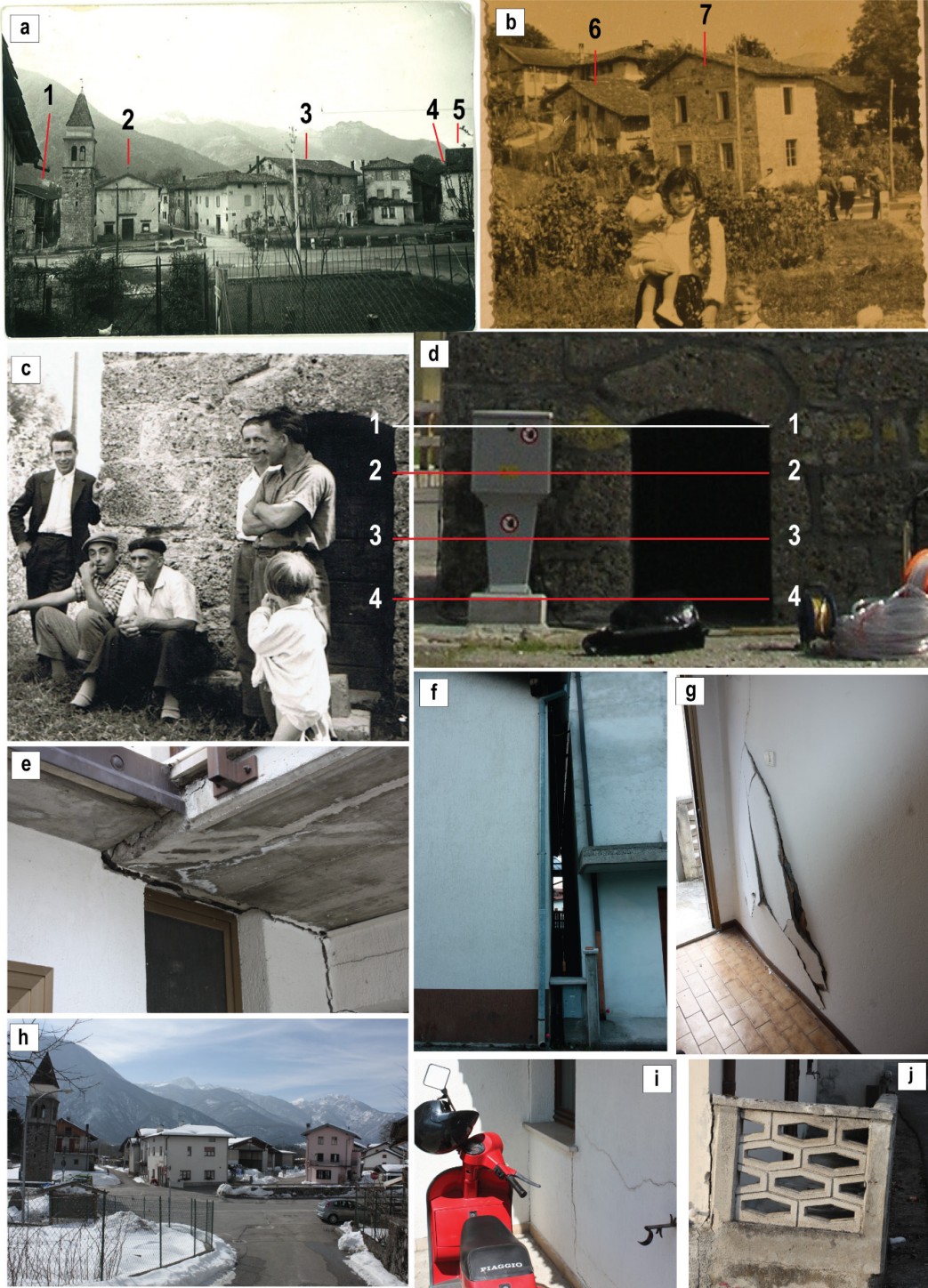

**Figure 11.** (**a,b**) Photos dated back to 1954, witnessing that the subsiding phenomena heavily affected the area in the past and their evolution has led to the demolition of some buildings; (**c,d**) the bell tower pictured in the sixties and nowadays, respectively. White lines join the same points on the two pictures, the red line highlights the subsidence of the structure; (**e,f,i**) report the fractures present on the walls and at the junction with another house noted as number 9 in Figure 10; picture f shows the convergence between two adjacent buildings; (**g,j**) damages on the internal wall of the house noted as 11 in Figure 10 and the external concrete fence is shown to be detached from the corner; (**h**) panoramic view where the bending of the bell tower and the white house is evident.

Thanks to the collaboration with the public administration and citizenship, several vintage pictures of the area were collected soon after Second World War, allowing the reconstruction of the housing settlement evolution [90]. Indeed, after the end of the war, several renewed buildings had to be demolished due to instability problems. Among these, the church, close to the bell tower (two in Figure 10b), the hayloft (one in Figure 10b), three buildings S of the main road and four to the N no longer exist (three, four and five and six and seven, respectively in Figure 10b). In the same area, other buildings are actually damaged (in red in Figure 10b). One of the most critical constructions is the bell tower. Comparing the photos of the 1960s with the current photos, it is possible to note the changes in inclination and its evident lowering (Figure 11c,d).

The building marked as nine in Figure 10 not only shows important cracks externally but also internally. When entering the house, the sensation of moving on an inclined plane is very impressive. The building is bending towards its neighbor (Figure 11f) and the inclination is so important that it is difficult to open or close some doors and windows.

The white house (Figure 11h) is bending and its normal use is compromised.

Overlapping the maps of the damaged buildings and the PSI surface displacement, it is possible to find a good correspondence. The only heavily damaged building, which seems to be stable according to the PSI data, is number 11 on Figure 10, but the many internal and external impressive cracks (Figure 11g,j) are irrefutable pieces of evidence of an important on-going movement, which is not apparent on PSI data, possibly because the building is linked with the neighboring building, which limits the overall downward movements. Therefore, the interferometric results were preliminary compared with information provided by the precise leveling campaign completed within the period from October 2012 to August 2015. The campaign was subdivided into nine different surveys performed by the topographical technical study [78]. Benchmarks were placed both at the bottom and at the top of selected buildings, which thus allowed confirmation of noticeable differential movements (Figure 12). In less than three years, data analyses of the bell tower showed a vertical lowering of 2.5 cm and a horizontal NW-displacement of more than 4 cm. The benchmark on building nine (Figure 10), in the same period, highlighted a progressive vertical lowering of about 2 cm and a horizontal E-displacement of 1.5 cm. The pink building (number 10 in Figure 10) showed an NNW horizontal displacement and a vertical lowering of about 2 cm. On the other hand, building number 11 in Figure 10, in three years, had irrelevant vertical and horizontal movements, possibly as a consequence of its connection with the neighboring building, as described above.

Analyzing the 3-year data and comparing the results obtained by combining the precise leveling technique and the interferometric approach, it emerges that the vertical lowering results are in the same order of magnitude and are almost similar, even if the two different techniques are totally independent from the other. In order to make a comparison, the horizontal displacement has been analyzed only in its E-W component (arrows in Figure 12).

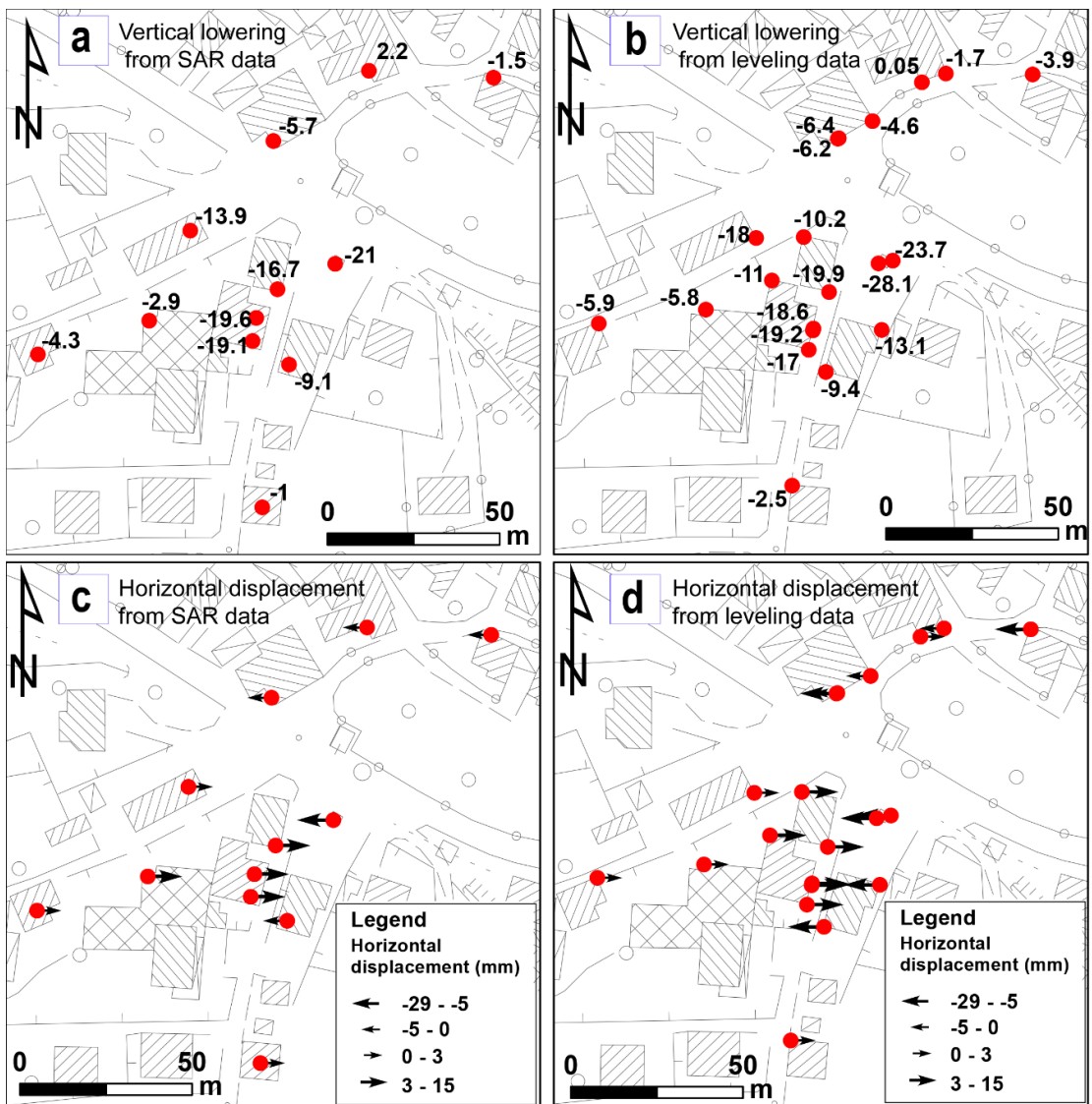

**Figure 12.** (**a**) Vertical lowering and (**c**) E-W horizontal displacement from interferometric data; (**b**) vertical lowering and (**d**) E-W horizontal displacement from precise leveling. All data are related to the period 2012–2015 and are in mm.

*4.3. 3D GPR*

In order to characterize the shallow subsurface in all the zones between the most damaged buildings and where the interferometric analysis showed the highest vertical and horizontal deformation rates, we collected a full 3D GPR dataset (Figure 6). At first, we analyzed single 2D profiles to understand if breaks and damages at the surface could be correlated with hidden structures. This correlation was apparent in several cases (e.g., Figure 13a); in fact, below some sub-horizontal reflectors attributable to the road pavements and its base, some sinking structures with different wideness were recognized (Figure 13). In this context, moving just a few meters apart, the shallow subsurface can be totally different with almost stable zones close to each other, characterized by severe sinking. As a consequence, the strong differential subsidence can emphasize the damage of the buildings. Unfortunately, by just considering 2D profiles, it is almost impossible to highlight the lateral limits of the sinkholes. In fact, even when the data are clear, it would be highly subjective to define which is the lateral border and its following areal path.

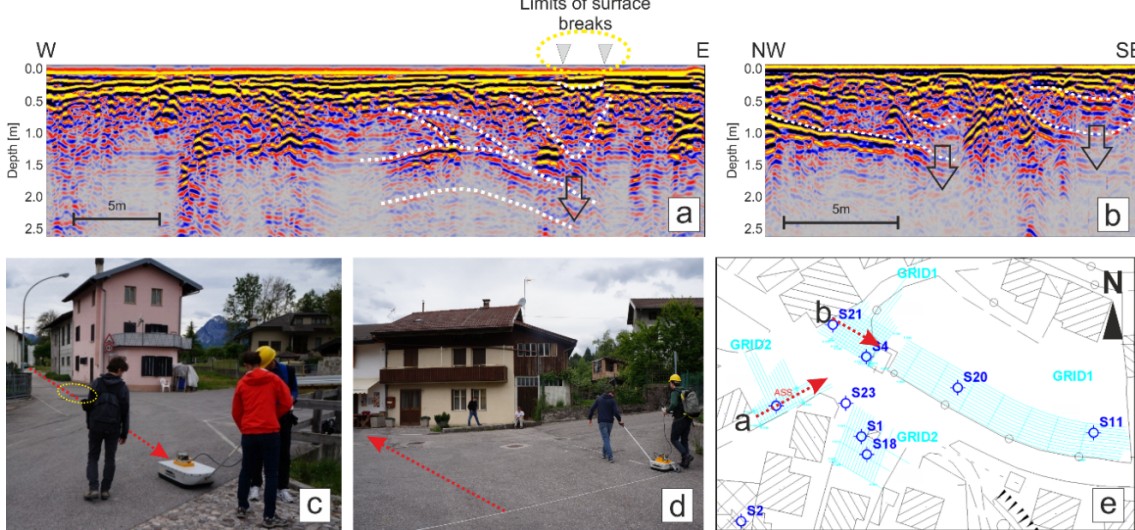

**Figure 13.** Exemplary 2D processed and interpreted 2D GPR profiles (**a**,**b**). On (**c**,**d**) photographs, the paths of profiles in (**a**,**b**) are highlighted, respectively. (**e**) Location map of the profiles in (**a**,**b**). Dotted white lines mark down-dipping layers related to the presence of sinkhole phenomena; black arrows are located where the deformation reaches the maximum; dotted yellow ellipses on (**a**,**c**) show the area where breaks at the surface are apparent.

We therefore considered the 3D depth volumes integrating 2D (in-line and crossline) visualization with depth slices and with any other arbitrary traced section (Figure 14).

The details of Figure 14b,c show that often the lateral continuity of the reflectors is low due to the pipes (P) and their related excavations or to local diffractions and reverberation effects of manholes. The analysis of amplitude or attribute depth slices can overcome such constraints, allowing the better imaging of the lateral limit of sinking areas, clearly discriminating the pipes and other coherent anthropic structures. Coherency attributes can further help this process as shown in the exemplary case of Figure 14a. In fact, while the original reflection amplitude has a range of values from negative to positive, coherency, as well as other attributes such as instantaneous or RMS amplitude, it can assume only positive values, making data interpretation and feature recognition easier and less subjective.

Furthermore, since the 3D dataset is georeferenced, it is straightforward to plot depth slices on topographic base maps, and/or integrating the geophysical data with any other information, including boreholes, displacement rates, geological data and so on.

Another interesting result is that even when several pipes and technological networks are present, the full 3D GPR dataset allows the highlighting and mapping of the sinking areas. As an example, in Figure 15, 0.50 m and 0.85 m depth slices are plotted on the topographic map of the study area. While at both depths several pipes are recognizable, two subsiding areas (labeled with letter D) are apparent and can be easily delimited. In detail, the northern one is not visible on the shallower slice because the large number of intersecting pipes probably obliterates such a feature, which is instead very clear on the deeper slice lying only 35 cm below the other one. We remark that during the GPR analysis and interpretation process, it is possible to continuously move from one slice to the other, which in the present case are separated by only about 1.5 cm, as a combined function of sampling interval and subsurface EM velocity.

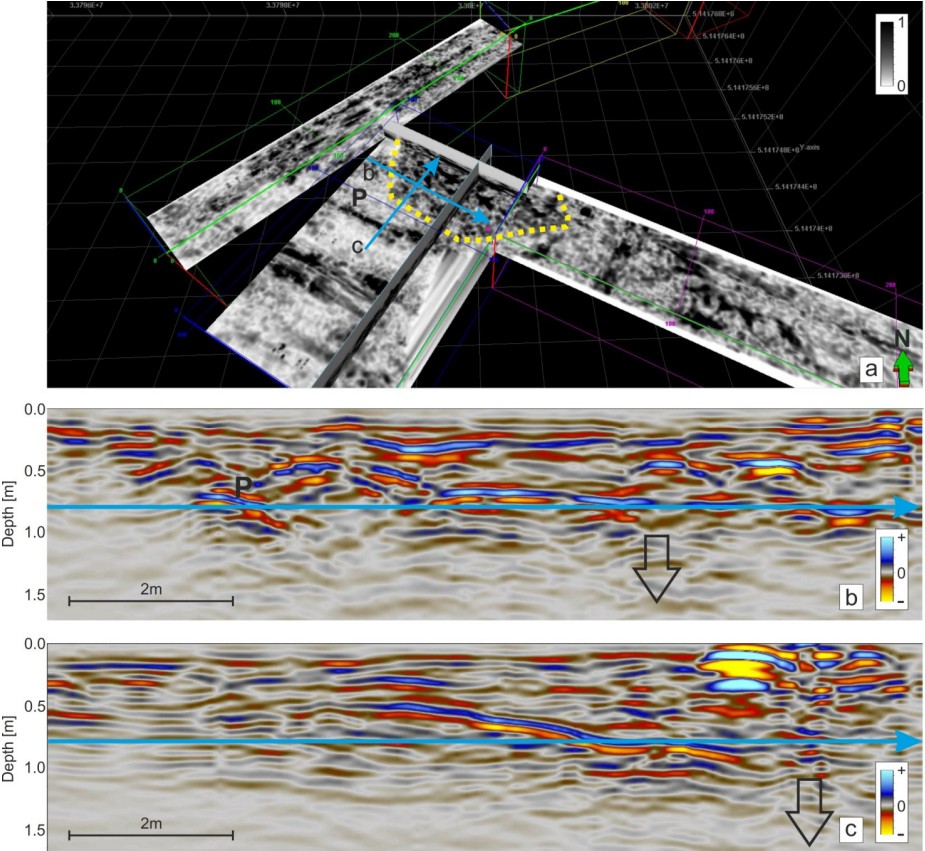

**Figure 14.** 3D GPR data analysis. (**a**) Perspective view of a portion of the GPR volumes with coherency depth slices at a depth equal to 0.8 m. The dotted yellow line marks the limit of a large depression, which is also imaged by the 2D perpendicular profiles (inline and crossline) in (**b**,**c**). Light blue lines show the location of shown profiles and the depth slice. The P label marks a clear pipe, while black arrows are located where the deformation reaches the maximum.

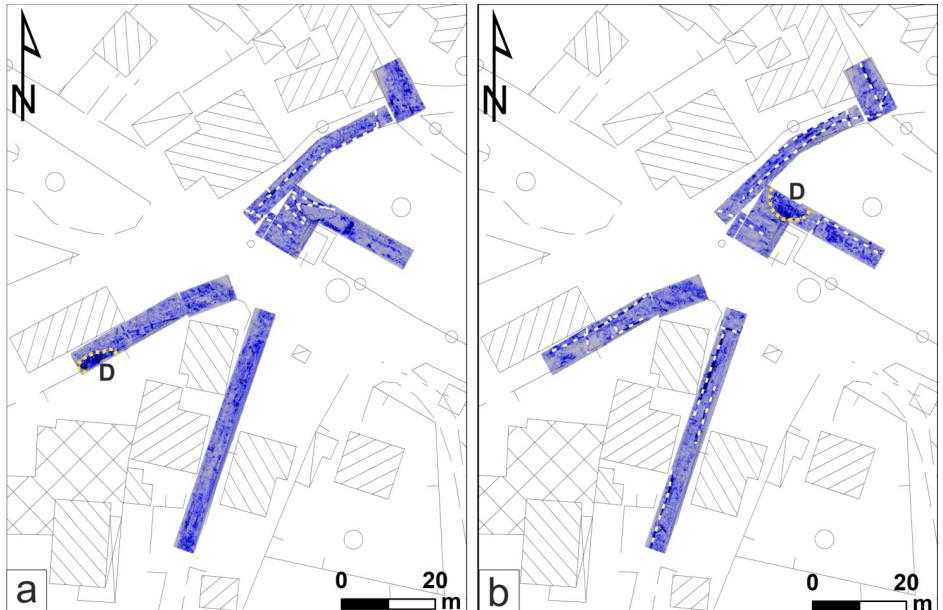

**Figure 15.** Exemplary 3D GPR depth slices at 0.50 (**a**) and 0.85 m (**b**). Yellow dashed lines mark main depocenters (D), while white lines mark some of the imaged pipes.

## 5. Discussion

The investigated area, despite being in a wide valley floor, has been known for instability problems since the 1940s. Due to the damage on the infrastructures, already visible in the past, several geological studies have been performed in the area, allowing the reconstruction of a preliminary geological model and to establish the phenomena causing such instabilities. However, being in a mantled karst environment, where roads and buildings cover the existing pieces of geomorphological evidence, the classical approaches or similar approaches, as proposed in the recent paper by Sevil et al. (2020) [6], do not allow very precise outlining and mapping of the single phenomenon, which involves specific infrastructures. The approach used in the area encompassed different multi-scale complementary methods, which allows the investigation of the subsurface of urbanized areas and represents both a novelty and a strength in the research being applied here for the first time. In particular, a multi-scale and multidisciplinary approach is essential to quantitatively evaluate and map the zones that are most vulnerable to sinking phenomena, while also highlighting areas that are still not affected by apparent deformations.

Thanks to reflection seismics, we highlighted the deepening of the evaporitic karstified bedrock towards S, which reaches the study area depths exceeding 60 m. Moreover, we imaged an irregular topography of the bedrock, probably linked to deep karst phenomena and the sinking accommodation by the surface sedimentary layers. This complex situation is not apparent on the surface, mainly due to the anthropic rehash and the presence of the buildings. At a totally different scale and resolution level, 3D GPR provides details about the shallow subsurface morphology showing local depocenters, which are not always linked to seismic reflections, due to the limited penetration depth of GPR technique.

With the PSI approach, we can extend the analyses identifying and mapping the areas where the geostatical criticisms are present to where further investigations should be focused on. From the study of the PSI data in the area of the Enemonzo municipality, emerged the presence of a zone where the velocities of deformation were higher (the Quinis area). Focusing on the latter, the data processing allowed us to be able to calculate the downward and horizontal displacements in the period between 2012 and 2016, identifying a portion of the territory where the sinking phenomenon is active and perceptible on the surface thanks to the damage to several buildings. For the interferometric analyses, we used the above-mentioned period because we had a set of precise leveling data of the damaged houses for the same time range available. Of course, it would be desirable to work with longer datasets, but the aim of this study is to compare results obtained using different sets of data collected with different methods, proposing an integrated multi-scale approach. Therefore, even with the two different analyses obtained with the independent PSI and leveling techniques, the obtained results are impressively coherent, validating the quality and the reliability of both methods. The spatial and temporal resolution of the deformation data acquired by the PSI are frequently not accurate enough for capturing active sinkholes of reduced size and/or characterized by catastrophic collapse. In this case, the characteristics of the movements allow the presented approach to be able to identify just the areas where the velocities are higher. Nonetheless, the use of satellite missions capable of acquiring radar data with much shorter revisit time and higher spatial resolution (such as COSMO-SkyMed) will significantly contribute to reduce these constraints, allowing the better identification of these kinds of phenomena. One of the limits of the leveling approach is that we can precisely define the movement of a single point, but to outline a moving area, it is necessary to test several points, which is a time- and logistically-consuming approach. With the PSI, it is instead possible to delineate the area involved in subsidence phenomena, especially in urban zones where the presence of targets such as buildings, monuments or pylons allows an excellent coverage of the persistent scatterers (PSs). Where PSs are not present, as in cultivated fields and in vegetated areas, the full 3D GPR method can be a helpful alternative. Usually, it is applied for investigations different to sinkhole detection, but this new method provides excellent results especially if used in flat and relatively wide areas, such as on roads or grasslands. The depth of investigation is usually not high (approximately maximum 2 m in

the present case), but the high resolution allowed the clear detection of two mantled subsiding features, which also fall in the area identified by the PSI data as a high sinking zone.

In an anthropized area, we found that local minor damages (such as breaks and fissures of the road pavement, sidewalks and containment walls) can also be directly related to local sinking effects (Figure 16). This is not always clear at a first glance or just by speaking with the citizens, because similar phenomena in other contexts are due to pipe breaks, seasonal freezing or dilatational effects, or are a consequence of roots growth and so on. Therefore, an integrated and multi-scale approach is mandatory in order to cross validate the results obtained by different techniques and observations and to highlight phenomena that cannot be made visible by using just one or a few methodologies.

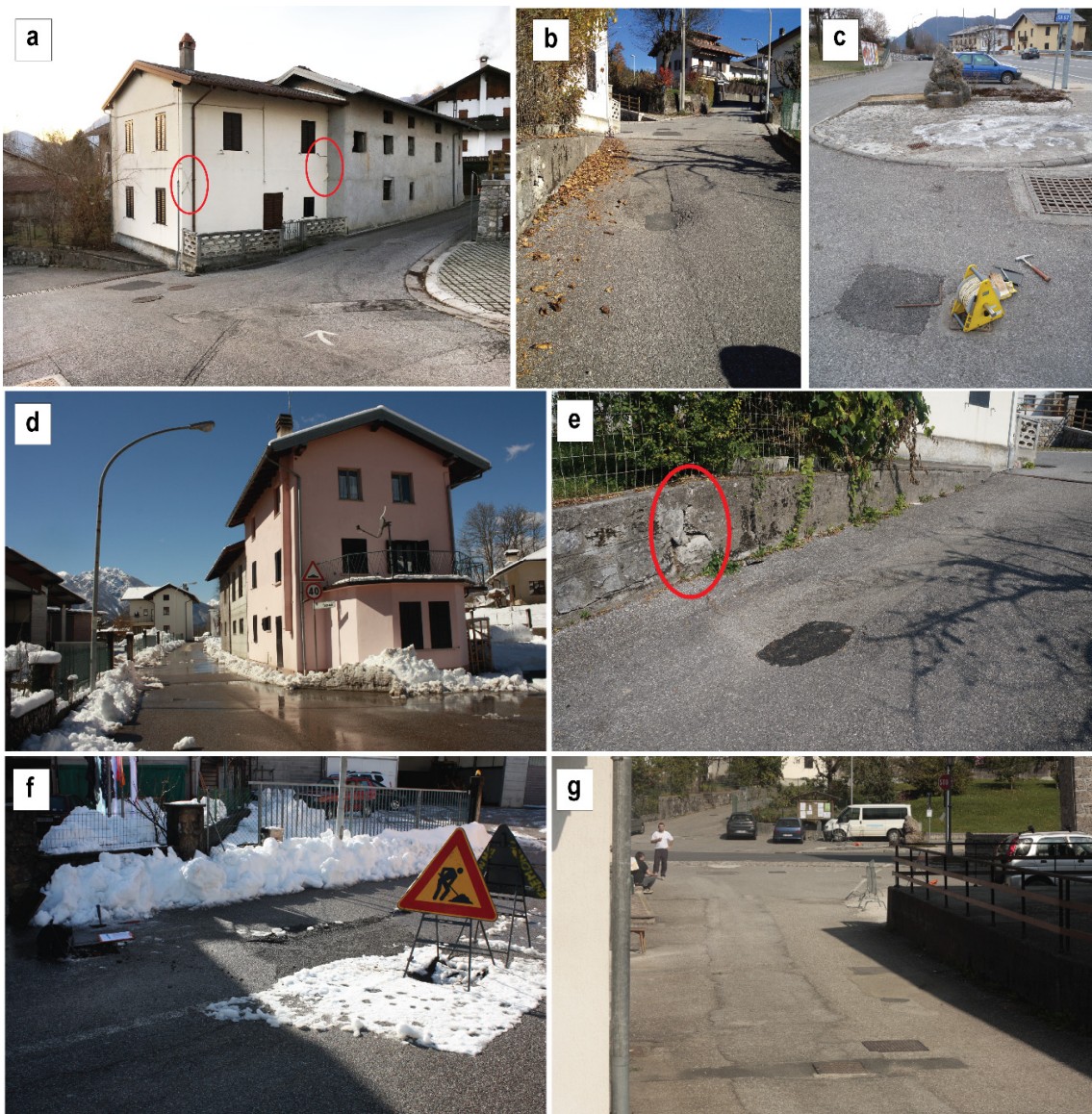

**Figure 16.** Collage of photos with evidence of breaks and fissures of the road pavement, sidewalks, and containment walls. (**a–g**) letters identify the damaged sites and have a corresponding position in Figure 17. (**a**) Red ellipses highlight the presence of strands reinforcing and linking the building; (**e**) cracks on the sidewall correspond to an active sinkhole.

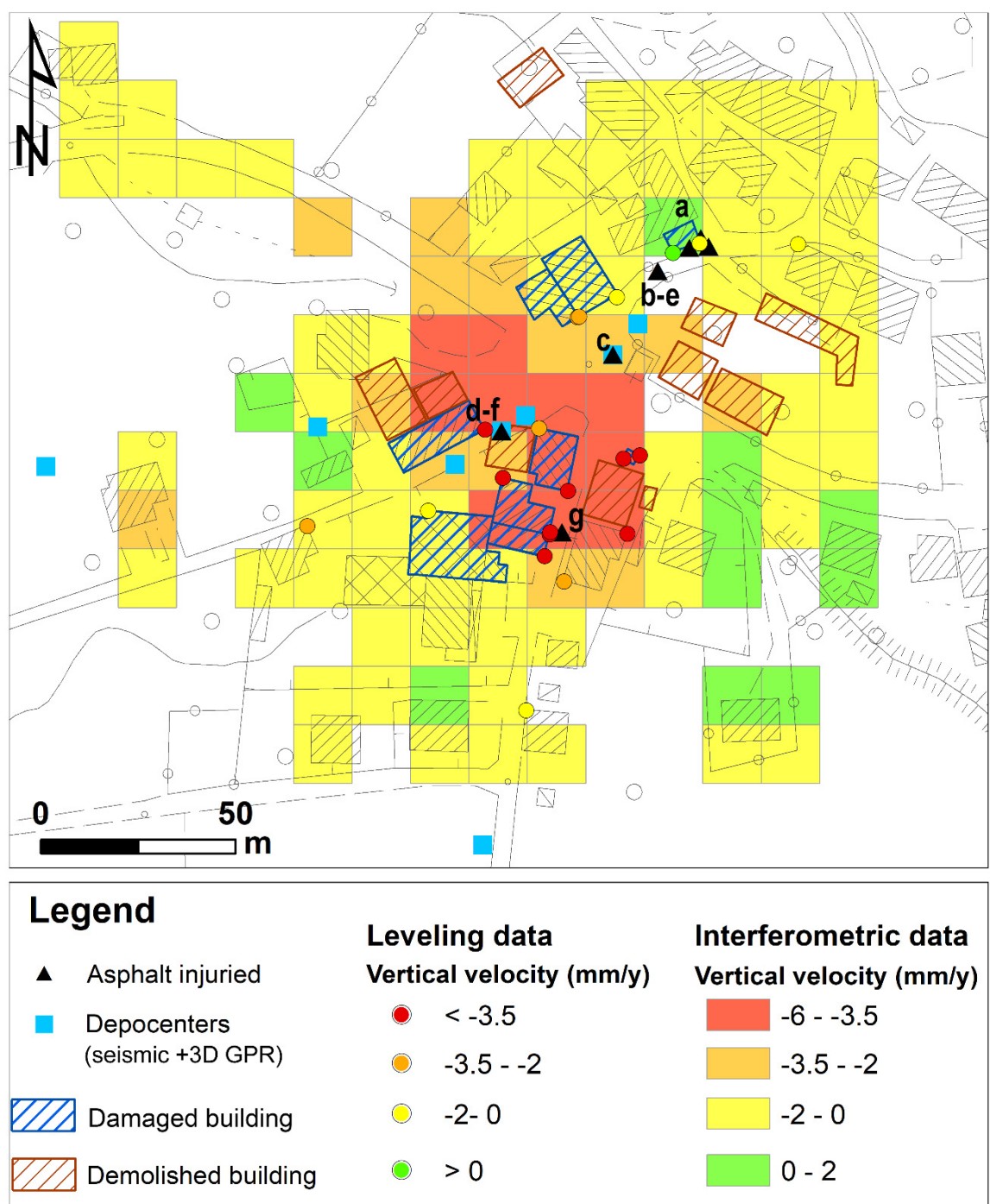

**Figure 17.** Synthetic map showing the combination of data obtained by the applied methodologies and highlighting the highest vulnerable zone. See text for details. Letters are related to damages presented in Figure 16.

The map shown in Figure 17 provides a synthesis of the obtained results. We combined geophysical, PSI and leveling information, while also locating the damaged buildings and local asphalt pavement breaks or renovation. Furthermore, we drew on the map the buildings that are nowadays demolished, by using vintage photographs and historical maps. The data are consistent, with the most relevant present damages and the demolished buildings within the zones with higher sinking velocity on the basis of both leveling and PSI. Geophysically imaged depocenters are sparser, but most of them lie

within the most critical area and perfectly correlate with the damage to the local pavement. A relevant exception and a still open issue is related to the house identified by number 11 in Figure 10b, which, as already pointed out, clearly presents signs of major damage, but seems to be in a stable area (from both leveling and PSI), where 3D GPR does not image any relevant feature except several shallow pipes (Figure 15). In front of this house, just below the asphalt, a void was present, but it was filled by loose material just before the 3D GPR acquisition, so it possibly masks all the sinking related features. In addition, the strands reinforcing this building and linking it to the house towards the west (see red ellipses in Figure 16a) probably limited the strains and differential settlements.

The lesson learned is that, especially in anthropized environments, it is mandatory to use different integrated techniques, without forgetting the role of the fieldwork of the geologists who can detect the precursors or the already occurred, even elusive, signs of ongoing or incipient sinking. In addition, it is essential when producing large scale maps to consider that local but dangerous and relevant phenomena can be masked by the resolution limits of the adopted methodology and/or by the anthropic rehash and existing infrastructures.

## 6. Conclusions

In a complex geological and hydrological framework, as in the study area, a multidisciplinary and multi-scale approach is mandatory to identify and map the zone most affected by sinking phenomena. While punctual data such as borehole stratigraphy, local groundwater level variations with time, extensometers measurements and geotechnical parameters are useful to highlight local hazards due to occurring deformation, the proposed integrated methodology addresses a complete and quantitative assessment of the vulnerability of the area.

Even if it is almost impossible to deterministically predict the occurrence of future sinkholes and their precise behavior in terms of both location and evolution with time, with the proposed integrated approach, we were able to:

(1) Image the most relevant subsurface features at a scale of tens of meters thanks to reflection seismics' profiles (checked and validated by borehole stratigraphy);

(2) Identify and map the zones with higher vertical movements thanks to the multi-year interferometric data analysis;

(3) Cross-checking both vertical and horizontal movements by integrating repeated precise leveling measures with interferometry;

(4) Highlight and map the main shallow depocenters thanks to full 3D GPR;

(5) Obtain a summary map showing the highest vulnerable zone as a function of deformation thresholds set on the basis of the observations and peculiarities of the area (Figure 17).

The latter result has not been common in the most recent similar studies. In fact, the current approach to obtain a hazard assessment for sinkholes of an area, even when multidisciplinary, is mainly focused on the visual comparison of aerial photographs and satellite images used to define the limits of one or a few single sinkholes with time, e.g., [6]. It is apparent that such an approach has an intrinsic high subjectivity level and leads to just qualitative results.

**Author Contributions:** Conceptualization, A.B. and C.C.; data curation, A.B., C.C., E.F., G.A. and A.M.; funding acquisition, C.C. and L.Z.; investigation, C.C., E.F., A.M. and L.Z.; methodology, A.B., C.C., E.F., G.A. and L.Z.; project administration, C.C. and L.Z.; supervision, C.C., E.F. and L.Z.; writing—original draft, A.B., C.C., E.F. and G.A.; writing—review and editing, C.C. and E.F. All authors have read and agreed to the published version of the manuscript.

**Funding:** This research was partially funded by the Geological Survey of the Friuli Venezia Giulia Region, in the framework of the following projects: (1) Realizzazione del censimento regionale dei sinkhole e relativo GIS con corso formativo e predisposizione delle linee guida di rilevamento ed informatizzazione nonché glossario dei termini utilizzati (prot.no. 801 del 28 October 2013, CUP: J92F16001310002); (2) accordo attuativo di collaborazione per la definizione e quantificazione della pericolosità dei sinkhole nei litotipi evaporitici Carniani dell'alta valle del Tagliamento (prot.no. 877 del 06 October 2016); (3) accordo attuativo di collaborazione per la misurazione delle geometrie e delle deformazioni indotte dalla presenza di sinkhole (prot.no.1254 del 21 November 2018); (4) progetto

di ricerca per la definizione del fenomeno di emissione gassosa in Comune di Enemonzo, località Quinis e delle eventuali correlazioni con i presenti fenomeni di sinkhole finalizzate alla valutazione della pericolosità associata nonché delle soluzioni tecniche di mitigazione o compensazione del dissesto (prot.no.286 del 12 June 2014, CIG: Z4C0C938BE); (5) accordo attuativo di collaborazione per l'aggiornamento censimento e pericolosità dei sinkhole del territorio regionale (prot.no.0035220 del 27 July 2020).

**Acknowledgments:** The authors would like to acknowledge the functionaries of the Geological Survey of FVG Region: Chiara Piano, Antonio Bratus and Franco Liuzzi as scientific coordinators of the joint different projects with the University of Trieste, Dept. of Mathematics and Geosciences and for providing interferometric data. We acknowledge the Italian Space Agency because the research was carried out using CSK® products © of the Italian Space Agency (ASI), delivered under a license to use by ASI being a product available for the FVG Region. We also acknowledge the service Rheticus® Displacement and Vincenzo Massimi and Sergio Samarelli from Planetek Italia as well as Raffaele Nutricato and Davide Oscar Nitti from Geophysical Applications Processing. We also acknowledge Schlumberger through the Petrel interpretation package and Halliburton through Seispace—ProMAX suite academic grants awarded to University of Trieste. We would like to thank Paolo Gabrielli for his help during seismic data acquisition and processing.

**Conflicts of Interest:** The authors declare no conflict of interest.

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
