# Peer review of "Non-Invasive Methodological Approach to Detect and Characterize High-Risk Sinkholes in Urban Cover Evaporite Karst: Integrated Reflection Seismics, PS-InSAR, Leveling, 3D-GPR and Ancillary Data. A NE Italian Case Study"

_remotesensing, doi:10.3390/rs12223814_

Round 1
Reviewer 1 Report
Dear authors,
I found this work both interesting and very well written. With respect to the manuscript structure, I am in opinion that manuscript is also well organized and balanced giving all necessary information in introduction, explanation of theoretical background, results, discussion and results. In overall proposed manuscript is very convincing and results are highly valuable.
The figures are in overall of good quality, beside Figure 4 that needs some minor improvements.
In overall, I am in opinion that reviewed manuscript requires very few corrections, and after these minor corrections I would recommend the manuscript for a publication.

Author Response
Answers to Reviewer 1
As Authors, we are pleased to know that the paper has been appreciated and we really thank the reviewer for the considerations and suggestions done which will improve the overall quality of the manuscript.
On the manuscript we made the suggested revisions which, in details, are listed below:
Line 25: The word “clearly” has been deleted as suggested.
Line 93: References were added [38, 39, 40, 41].
Line 102: The word “Earth” had been deleted as suggested.
Line 122: We add the Figure 3 reference, as suggested.
Line 137-138: We verified the space line which has been corrected.
Line 141: We did not modify the sentence because the thickness of the Quaternary deposits is specified in the lines 144-145 as follows: “to the North, it is approximately a few meters and deepens moving toward South up to more than 60 m in correspondence of the Tagliamento river terraces”.
Figure 4: We modified the figure according to the suggestions by thickening the fault lines and adding the symbols for the available bedding strike/dip data.
Line 239: We added Table 2 to better visually highlight the processing flow, as suggested.
Line 517: We changed forties’ into 1940s.
Line 599: We adjusted the Figure’s size in order to let the figure capture to be just below the figure itself.

Reviewer 2 Report
Dear Authors,
The subject was well described in the abstract of the MS. Although the first chapter introduces well the monitoring methods according to sinkholes occurrence detection, there is a lack of detailed information about countries facing the issue of sinkholes. Some methods and countries around the World have been summarized at the fig.2, but there are some other countries which has problems with the sinkholes and their early detection as well (e.g., Iran, Poland, Great Britain). They are not shown on the World map, but perhaps shall be.
The monitoring methods have been introducing in the research with increasing resolution level. There were reflection seismic, DInSAR, 3D GPR and levelling. Finally, the whole research method results have been integrated on the map. This result in the form of cartographic feature can be found as interesting and well designed. The proposed research methodology was proved on the case study of small village. The only weakness in the research part of the MS would be lack of the accuracy discussion according to the research methods. The discussion chapter clearly presents the interrelationships between the applied research methods. The outcomes are clear and comes directly from the research results.
The particular remarks were:
- at the Fig. 12, the description is confusing and awkward.
Although the research study is interesting and clearly present, the research niche was not shown clearly. That is why the manuscript can only be seen as a case study. In my opinion this is not enough for publishing in the RS in the present form. I would propose to improve the MS by proposing the research niche, more extensive introductory part and the accuracy discussion.
Author Response
Answer to Reviewer 2
Dear Authors,
The subject was well described in the abstract of the MS. Although the first chapter introduces well the monitoring methods according to sinkholes occurrence detection, there is a lack of detailed information about countries facing the issue of sinkholes. Some methods and countries around the World have been summarized at the fig.2, but there are some other countries which has problems with the sinkholes and their early detection as well (e.g., Iran, Poland, Great Britain). They are not shown on the World map, but perhaps shall be.
As Authors, we are pleased to know that the research has been appreciated and we thank the reviewer for the suggestions which will improve the overall quality of the manuscript.
Figure 2 Thank you for your suggestion. Figure does not represent all the countries affected by sinkholes, which are definitively higher in number. It rather represents the sites in which specific remote sensing techniques were applied, especially SAR. We agree indeed with the reviewer who spur us to improve our work adding new references. We modified the map in Figure 2 according to the given suggestions and modified the figure caption in order not to mislead the reader from figure understanding.
The monitoring methods have been introducing in the research with increasing resolution level. There were reflection seismic, DInSAR, 3D GPR and levelling. Finally, the whole research method results have been integrated on the map. This result in the form of cartographic feature can be found as interesting and well designed. The proposed research methodology was proved on the case study of small village. The only weakness in the research part of the MS would be lack of the accuracy discussion according to the research methods. The discussion chapter clearly presents the interrelationships between the applied research methods. The outcomes are clear and comes directly from the research results.
The particular remarks were:
at the Fig. 12, the description is confusing and awkward.
As suggested also by reviewer 3, we modified Figure 12 in order to make it clearer.
Although the research study is interesting and clearly present, the research niche was not shown clearly. That is why the manuscript can only be seen as a case study. In my opinion this is not enough for publishing in the RS in the present form. I would propose to improve the MS by proposing the research niche, more extensive introductory part and the accuracy discussion.
Regarding the Introduction, as suggested, we added the references concerning sinkhole analyses done in Iran, Poland, and Great Britain ([57] Malinowska, A.A.; Witkowski, W.T., Hejmanowski, R.; Chang, L.; van Leijen, F.J.; Hanssen, R.F. Sinkhole occurrence monitoring over shallow abandoned coal mines with satellite-based persistent scatterer interferometry. Eng. Geol. 2019, 262, 105336. https://doi.org/10.1016/j.enggeo.2019.105336.
[58] Vajedian, S.; Motagh, M. Extracting sinkhole features from time-series of TerraSAR-X/TanDEM-X data. ISPRS J. Photogramm. Remote Sens. 2019, 150, 274-284. https://doi.org/10.1016/j.isprsjprs.2019.02.016
[59] Novellino, A.; Cigna, F.; Brahmi, M.; Sowter, A.; Bateson, L.; Marsh, S. Assessing the feasibility of a national InSAR ground deformation map of Great Britain with Sentinel-1. Geosciences 2017, 7(2), 19. https://doi.org/10.3390/geosciences7020019).
Regarding the discussion, the chapter has been implemented according to the suggestions given also by reviewer 1 and reviewer 3), Lines 111-112.
In the presented research, the authors analyze the Alta Val Tagliamento Valley, focusing on the Quinis area. Even if the investigated area is small, it is representative of a geological and structural context typical and diffuse in the Alpine and Appenine areas and, more generally in other parts of the world. The investigations here realized and the obtained results could be easily applied to every other context in the world where the environment is characterized by the presence of an evaporite bedrock mantled by Quaternary deposits (a situation occurring for example in some parts of Iran, Iraq, Spain, Arabian Peninsula, among the other countries). Furthermore, we think that our paper is not only a case study, but rather it represents an useful methodological approach, as also highlighted for instance by reviewers 1 and 3.

Reviewer 3 Report
Dear authors,
I reviewed the paper entitled “Non-invasive methodological approach to detect and characterize high-risk sinkholes in urban cover evaporite karst: integrated reflection seismics, 3D GPR, leveling, DInSAR and ancillary data. A NE Italian case study” by Alce Busetti et al.
The paper deals with an integrated methodology using different geophysical and geodetic (spatial and ground based) techniques as seismics reflection, satellite radar interferometry, leveling and 3D-GPR to assess the detection of sinkholes in a small village in NE Italy. This is a nice application that places value on the combined use of these different techniques.
The paper is well written. The introduction exposes clearly the background and objectives of the study. Please, add some references for seismic reflection applied to sinkhole detection. The study area is well described. The data and methodology are well exposed although I have some comments about it below, and the results show with good detail and graphical documentation the outcome of the experiments with the different techniques. In the discussion I miss the comparison and discussion of the results with other similar studies/experiments. This would enrich this section a lot. Finally, the conclusions are supported with the results.
I suggest to maintain the same order when naming the techniques used in the paper, that is, in the title, abstract, and along the paper. For example, the order used in the methodology section.
Related to the satellite radar interferometry technique, I would not use DInSAR term in this case because DInSAR means to use “Differential” InSAR for getting a deformation map but it has a connotation of using only two images, before and after a deformation event. When you use a temporal series analysis you are using what it is called Advanced DInSAR techniques or Multi-temporal InSAR tecniques, which are usually named as A-DInSAR, MT-InSAR, etc. Two families of approaches using in these multi-temporal techniques are the PSI or PS-InSAR (a single master approach) to get Persistent or Permanent Scatterers (PS) and SBAS to get Distributed Scatterers (DS). In this paper the authors used the first familiy of multi-temporal techniques so I suggest to change the name DInSAR in this paper for PSI (Persistent Scatterers Interferometry), PS-InSAR, MT-InSAR, A-DInSAR or similar along the paper.
Is there any piezometer available in the area? It would be interesting to see the water level variation along the period of study and contrast it with the radar time-series.
Related to the unit of velocity, please, revise the paper (including figures) and unify the symbol used for it. You are using mm/y, mm/year and mm/years.
Replace “meters” with “m”.
In Figure 1, add a frame with geographical coordinates in plot b.
Figure 4 is cited before than Figure 3.
Line 189. Rewrite the sentence as: “In 6 houses important damages have been…”
Line 271 and equation (1). The component you get is East-West direction.
Line 275. Add “it” before “was computed…”
Section 3.2. How many images did you use for the ascending and descending datasets? Which are the date of these images? Which kind of CSK SAR images did you use and which is the ground resolution? Where is located the reference area for the processing and why? Please, add two plots with the temporal vs. perpendicular baselines distribution of the ascending and descending stacks.
Section 3.3. Add the dates of each leveling monitoring campaign.
Lines 298-299. Replace “TRIMBLE 5601 robot and TRIMBLE S6 instrument” with “TRIMBLE 5601 and TRIMBLE S6 robotic total stations”.
Line 333. Replace “analysing” with “analyzing” as you are using American English.
Line 371. “seen that” sounds strange. “seeing that” instead?
Figure 11d. What do the four lines represent? Levels of subsidence in some years? Which years?
Figure 12. Add a title to each subplot to improve the understanding.
Line 464. Replace “analysed” with “analyzed”.
Lines 565-567. The sentence is not clear. There are two “are” and a connector seems to be missing.
Line 590. Add “a” to “Figure 16”-> 16a.
Author Response
Answers to Reviewer 3
In black the reviewer suggestions/requests, in bold the authors’ answers.
Dear authors,
I reviewed the paper entitled “Non-invasive methodological approach to detect and characterize high-risk sinkholes in urban cover evaporite karst: integrated reflection seismics, 3D GPR, leveling, DInSAR and ancillary data. A NE Italian case study” by Alce Busetti et al.
The paper deals with an integrated methodology using different geophysical and geodetic (spatial and ground based) techniques as seismics reflection, satellite radar interferometry, leveling and 3D-GPR to assess the detection of sinkholes in a small village in NE Italy. This is a nice application that places value on the combined use of these different techniques.
The paper is well written. The introduction exposes clearly the background and objectives of the study. Please, add some references for seismic reflection applied to sinkhole detection.
We add a new reference in the Methods section for seismics reflection [66] as suggested. Please notice that just a few lines below there were 3 other specific references.
The study area is well described. The data and methodology are well exposed although I have some comments about it below, and the results show with good detail and graphical documentation the outcome of the experiments with the different techniques. In the discussion I miss the comparison and discussion of the results with other similar studies/experiments. This would enrich this section a lot. Finally, the conclusions are supported with the results.
As suggested, we add few sentences on the original manuscript highlighting the novelty of the proposed methodological approach even compared to papers recently published on the similar topic (Sevil et al., 2020).
I suggest to maintain the same order when naming the techniques used in the paper, that is, in the title, abstract, and along the paper. For example, the order used in the methodology section.
According to what suggested we modified the title following the order we used in the Methods and Results sections.
Related to the satellite radar interferometry technique, I would not use DInSAR term in this case because DInSAR means to use “Differential” InSAR for getting a deformation map but it has a connotation of using only two images, before and after a deformation event. When you use a temporal series analysis you are using what it is called Advanced DInSAR techniques or Multi-temporal InSAR tecniques, which are usually named as A-DInSAR, MT-InSAR, etc. Two families of approaches using in these multi-temporal techniques are the PSI or PS-InSAR (a single master approach) to get Persistent or Permanent Scatterers (PS) and SBAS to get Distributed Scatterers (DS). In this paper the authors used the first familiy of multi-temporal techniques so I suggest to change the name DInSAR in this paper for PSI (Persistent Scatterers Interferometry), PS-InSAR, MT-InSAR, A-DInSAR or similar along the paper.
As suggested we substituted “DInsar” with “PS-Insar (PSI).
Is there any piezometer available in the area? It would be interesting to see the water level variation along the period of study and contrast it with the radar time-series.
The suggestion is very interesting.
Yes, the are 20 piezometers in the area which have been monitored for several years using CTD divers which collect data (EC, T, water level) every 30 minutes (Zini et al. 2015, Calligaris et al. 2019). As reported in the text of our new paper, “The aquifer system is very complex and the water table has a rapid response to rainfall, in particular the water level fluctuations are large, with oscillations between 6 and 32 m. One of the main rainfall events recorded 248 mm of rain in 56 h (24-26 December 2013) with a consequent water table rise of 10 m and a maximum recorded velocity of 40 cm/h [18] witnessing a fast circulation of the groundwaters”.
SAR data have a temporal acquisition sampling of approximately 15 days, during which there could be important changes in the water level. This do not allow to compare the hydrogeological data that we have with the SAR data. To compare the data and verify if there are actual correlations between the two datasets we should have closer temporal sampling for SAR acquisitions, which we would realize in the next future.
Related to the unit of velocity, please, revise the paper (including figures) and unify the symbol used for it. You are using mm/y, mm/year and mm/years.
We modified the units in the text choosing mm/y as suggested and verified it also in all the figures.
Replace “meters” with “m”.
Where necessary we modified “meters” substituting it with “m”.
In Figure 1, add a frame with geographical coordinates in plot b.
We add a frame with geographical coordinates in plot b-Figure 1.
Figure 4 is cited before than Figure 3.
We add a figure reference at line 122 in order to modify the order of appearance of the figures, as also suggested by reviewer 1.
Line 189. Rewrite the sentence as: “In 6 houses important damages have been…”
We rewrote the sentence as follows “Actually six houses have been relocated due to the important suffered damages.”
Line 271 and equation (1). The component you get is East-West direction.
As suggested we modified it in line 277 and 285.
Line 275. Add “it” before “was computed…”
We added “it” to the sentence (line 281).
Section 3.2. How many images did you use for the ascending and descending datasets?
78 images in the ascending and 78 images in the descending datasets were initially used by Planetek to provide us the data.
Which are the date of these images?
Approximately an acquisition every 15 days. The used dataset 3a_A in ascending orbit and the 03a_D in descending orbit span the period 18/02/2012 - 08/09/2016 and 01/01/2012 – 08/09/2016, respectively.
Which kind of CSK SAR images did you use and which is the ground resolution?
Planetek used a single-look complex (SLC) images acquired in STRIPMAP/HIMAGE mode. The ground resolution is 3m.
Where is located the reference area for the processing and why?
For every landslide phenomenon studied by Planetek for the FVG region, the coordinates of the interest area and an extraction mask were defined. To these polygons a buffer of 500m was applied in order to enlarge the investigated area. As described in the report that Planetek provided, the reference point for the processing has been chosen by analyzing the characteristics of the detected PSs in the area of interest. Firstly, a common velocity value has been added to all the PSs. This common value refers to the velocity of a known reference point or, in case of absence, the reference point has been chosen in order to maximize the number of PSs with null (zero) velocity. Then, by analyzing the trend of the PSs with null velocity, a PS with a high coherence trend or low noise level can be eventually chosen as reference point.
Please, add two plots with the temporal vs. perpendicular baselines distribution of the ascending and descending stacks.
Sorry, but we do not have this information. We tried to contact Planetek, but we did not receive any answer in time.
Section 3.3. Add the dates of each leveling monitoring campaign.
We added the dates in brackets.
Lines 298-299. Replace “TRIMBLE 5601 robot and TRIMBLE S6 instrument” with “TRIMBLE 5601 and TRIMBLE S6 robotic total stations”.
We modified the sentence according the given suggestion: TRIMBLE 5601 and TRIMBLE S6 robotic total stations,…
Line 333. Replace “analysing” with “analyzing” as you are using American English.
We modified the word.
Line 371. “seen that” sounds strange. “seeing that” instead?
We modified as suggested.
Figure 11d. What do the four lines represent? Levels of subsidence in some years? Which years?
Pictures 11c and 11d refer to the same structure: the bell tower. Lines 1, 2 and 3 join same points on the two pictures. Picture c was taken in the sixties, while picture d has been collected this year (2020). Line 4 highlights the subsidence of the structure in between sixties and nowadays.
We modified the figure caption as follows: “White lines join same points on the two pictures, red line highlight the subsidence of the structure.” Accordingly, we modified also the Figure.
Figure 12. Add a title to each subplot to improve the understanding.
We added a title to each subplot as suggested.
Line 464. Replace “analysed” with “analyzed”.
We modified the word as suggested.
Lines 565-567. The sentence is not clear. There are two “are” and a connector seems to be missing.
We add a connector as suggested. Something was missed in the original version.
Line 590. Add “a” to “Figure 16”-> 16a.
Thank you for your suggestion: we did it.
